# Global burden of major chronic respiratory diseases among older adults aged 55 and above from 1990 to 2021: Changes, challenges, and predictions amid the pandemic

Anran Xu[1]◉, Yinqin Liu[1]◉, Shaobin Li[1], Chengyan Zhan[1], Yanqi Cheng[1], Chen Zhang[1], Hong Fang[1]*, Donghua Zhou[2]

1 Department of Traditional Chinese Medicine Preventive Health Care, Longhua Hospital Affiliated to Shanghai University of Traditional Chinese Medicine, Shanghai, China, 2 Pneumology Department, Fengxian Branch of Longhua Hospital, Affiliated to Shanghai University of Traditional Chinese Medicine, Shanghai, China

◉ These authors contributed equally to this work.
* 15000297742@163.com

## Abstract

### Objective

To characterize sex- and age-specific changes in the comprehensive burden of major chronic respiratory diseases (CRDs) and their attributable risk factors among adults aged ≥55 years globally, regionally, and nationally from 1990 to 2021 using the Global Burden of Disease (GBD) 2021 database.

### Methods

Utilizing the GBD 2021 database, we performed in-depth analyses and preliminary projections of global, regional, and national burden trends for chronic obstructive pulmonary disease (COPD), asthma, and interstitial lung disease & pulmonary sarcoidosis (ILD&PS) through multi-model approaches including but not limited to Age-Period-Cohort (APC) models, Joinpoint regression, and Bayesian Age-Period-Cohort (BAPC) modeling.

### Results

The overall global CRD burden among adults ≥55 years declined from 1990 to 2021. However, the Corona Virus Disease 2019 (COVID-19) pandemic differentially altered asthma and COPD prevalence and incidence trends globally: low Socio-demographic Index (SDI) regions experienced an accelerated increase in prevalence, while high SDI regions showed a steeper rise in incidence. High mortality and disability-adjusted life years (DALYs) rates remained concentrated in low-middle SDI regions, notably Asia and North America. Consequently, prevalent CRD cases in this age group

**Data availability statement:** **AT FTC: follow-up to ensure OSF repository is made public.** The data utilized in this study were sourced from the open-access Global Burden of Disease (GBD) database. GBD 2021 data are publicly available online (http://ghdx.healthdata.org/gbd-results-tool). Additionally, the raw data, analytical code, and supplementary materials associated with this study are accessible in Supplementary Material 2 and via the Open Science Framework (OSF) platform (https://osf.io/4aq67/?view_only=0db1dec18bc04d4eb-c4ae2ee29f8d098).

**Funding:** The publication of this study was supported by the Preventive Treatment of Disease in Traditional Chinese Medicine Specialist Alliance Construction Project (Grant No. 2024LM01 to Dr. Hong Fang) of Longhua Hospital affiliated with Shanghai University of Traditional Chinese Medicine.

**Competing interests:** The authors have declared that no competing interests exist.

reached 223 million (95% UI 206.5–241.5) in 2021—accounting for half of all-age cases—with 18.47 million incident cases (95% UI 16.97–20.11), causing 4.15 million deaths (95% UI 3.76–4.58) and 83.67 million DALYs (95% UI 77.49–90.36). Air pollution, smoking, obesity, and chronic cold exposure persistently influenced COPD and asthma prevalence across regions and sexes.

## Conclusion

The pandemic shifted global CRD burden trends, particularly for asthma followed by COPD. Concurrent with global aging, burden trajectories across SDI levels raise concerns. As COVID-19 becomes endemic, older adults will experience impacts from recurrent viral infections, increasingly manifesting in coming years.

## Introduction

In the past three decades, the global burden of chronic respiratory diseases (CRDs) has never truly been under control. Under the impact of the global Corona Virus Disease 2019 (COVID-19) pandemic, the overall disease burden, including CRDs, has reversed and increased [1]. Although previous studies have shown a downward trend in the overall prevalence of CRDs [2], the affected population remains large, and the disability-adjusted life years (DALYs) continue to rise [1]. In the 2019 United Nations report on "World Population Prospects," it was estimated that over the next 30 years, and even by the end of the century, the global population will continue to grow to 9.7 billion people, ultimately reaching around 11 billion. The age group with the fastest-growing population is those aged 65 and above [3]. Given such a large population and changes in age structure, the absolute burden of CRDs among middle-aged and elderly individuals will continue to increase throughout this century. In 2019, CRDs were the third leading cause of death globally [2], and this situation remained unchanged in 2021 after the impact of the pandemic [4]. The economic burden caused by CRDs has had varying effects across different regions of the world [5,6].

As the global population rapidly ages [7–9], the burden of CRDs will gradually increase. Therefore, this study focuses on the main CRDs that are strongly correlated with age and environment, which have broader public health significance [2,10]. These include age-related chronic obstructive pulmonary disease (COPD), interstitial lung disease and pulmonary nodules (ILD & PS), and asthma, which is influenced by environmental changes.

The arrival of the COVID-19 pandemic has had a profound impact on the lung health of older adults globally. On one hand, this impact is mainly reflected in the higher COVID-19 infection rate among older adults with pre-existing CRDs [11], and their continued decline in lung function after recovery from COVID-19, which makes them more susceptible to acute exacerbations of their existing CRDs. On the other hand, with the normalization of the pandemic, the regional spread of new variants continues to cause persistent lung damage to healthy older adults [12,13], increasing the likelihood of developing COPD and ILD & PS [14,15].

Additionally, after multiple COVID-19 infections, the risk of acute asthma exacerbations significantly increases [16]. In 2015, the United Nations General Assembly proposed "reducing premature deaths from non-communicable diseases by one-third through prevention and treatment" [17]. However, under the impact of public health emergencies, the global goal of controlling the progression of CRDs has become more critical than ever before and requires long-term attention.

This study is the first to analyze the global burden of CRDs in individuals aged 55 and above. We used the 2021 Global Burden of Disease database (GBD) and combined it with the Sociodemographic Index (SDI), to provide an overall description, risk factor analysis, and regional inequality research on three types of CRDs, from a global scale down to specific countries, while also examining trends in disease changes over the next few years. This research, which assesses a specific population, has significant practical relevance and aligns with the United Nations' Sustainable Development Goals, helping healthcare professionals and policymakers formulate targeted policies that can benefit a larger number of middle-aged and elderly individuals.

## Methods

### Overview

The GBD database is an international, large-scale collaborative project. It uses advanced modeling techniques, including the Cause of Death Ensemble modeling (CODEm) and the Bayesian meta-regression tool DisMod-MR 2.1, to conduct stratified analysis of disease burden, and has been widely validated in previous studies [1]. GBD 2021 provides multi-dimensional disease burden data for countries, regions, and the global level, including prevalence, incidence, mortality, years lived with disability (DALYs), years of life lost (YLLs), and years lived with disability (YLDs), covering 371 diseases and injuries as well as 88 risk factors [18]. Additionally, GBD 2021 includes COVID-19 as an infectious disease for the first time, estimating its impact on the burden of specific diseases [1]. All GBD 2021 data are publicly available online (http://ghdx.healthdata.org/gbd-results-tool). The estimation process of GBD can be referenced in previous reports or supplementary files [1,7](S1 File). Finally, when de-identified data is used in the GBD study, informed consent is not required, and this ethical guideline has been approved by the Institutional Review Board of the University of Washington. Furthermore, the raw data and relevant code employed in this research have been deposited on the Open Science Framework (OSF) platform.

### Data sources

We obtained three key types of data from the GBD 2021 database, including the disease burden of CRDs covered in this study, SDI data, and relevant risk factor data for CRDs. In addition, we extracted population data from 1990 to 2021 from GBD [19] and obtained corresponding ratios for the World Standard Population, as well as population projections until the end of this century, from the World Health Organization (WHO). These data help us comprehensively assess the global disease burden of CRDs. The subjects of this study are adults aged 55 and above, covering the period from 1990 to 2021. Additionally, the data collection and analysis in this study followed the Strengthening the Reporting of Observational Studies in Epidemiology (STROBE) guidelines.

### Case definition

The GBD 2021 database classifies diseases and injuries into four levels, with COPD, asthma, and ILD & PS classified as level four causes, while CRDs are classified as level three causes. CRDs are defined according to the Global Initiative for Chronic Obstructive Lung Disease (GOLD) global initiative, the Global Initiative for Asthma (GINA) asthma management guidelines, and the American Thoracic Society standards, and are coded using the International Classification of Diseases (ICD-10 and ICD-9) [2,6,20]. Detailed information is provided in the supplementary file (S1 File).

### Descriptive statistics

We used descriptive statistical analysis to assess the burden of CRDs across different global regions, visualizing and comparing the trends and distribution of CRD burdens over time, by age, or by gender, in conjunction with the SDI of each region. Additionally, we used the estimated annual percentage change (EAPC) to describe the overall changes in the burden of CRDs across different regions and countries from 1990–2021 and 2019–2021.

### Trend analysis

Analyzing the changes in disease burden and explaining the driving factors is a key part of this study. We first used EAPC to describe the linear trends for all regions and countries [21]. However, this model may obscure important phase transitions. Therefore, we used Joinpoint regression to automatically detect significant inflection points and calculated the annual percentage change or average annual percentage change (AAPC) for each phase. Compared to traditional models, this method is more sensitive to phase trend changes, such as those before and after the COVID-19 pandemic, making it an ideal tool for assessing disease burden changes over decades [22]. Additionally, we used the NIH APC Web Tool (based on the Intrinsic Estimator, IE method) [23]to assess the age, period, and cohort effects on global data from 1992–2021, and to satisfy the linear dependency relationship $cohort = period - age$. The IE method resolves the issue of unidentifiability through minimum norm constraints, allowing us to simultaneously assess the impact of the three-dimensional time factors on disease burden. The Age-Period-Cohort (APC) model has been widely validated and used in previous trend analyses [23]. Finally, to further analyze the impact of population aging on disease, we conducted a decomposition analysis [24], which described global disease burden differences from three perspectives: population growth, population aging, and epidemiological changes.

### Cross-National Inequality and Risk Factor Analysis

Following WHO recommendations, we used the Slope Index of Inequality (SII) and the Concentration Index (CI) to assess the distributional inequality of disease burden. The SII captures average health level differences between populations, while the CI measures the extent of health inequity across countries with different socioeconomic statuses. Additionally, we downloaded risk factor data for CRDs from GBD. GBD 2021 adopts the Comparative Risk Assessment (CRA) framework to quantify the contribution of various risk factors to disease burden through the "exposure-risk-attribution" chain. This approach ensures comparability across regions and provides policymakers with quantitative evidence of the "avoidable burden" caused by each risk factor.

### Disease burden forecast

We first used the Bayesian Age-Period-Cohort (BAPC) model as the primary forecasting method to predict changes in age-standardized prevalence and incidence of CRDs over the next decade. This model uses Integrated Nested Laplace Approximation (INLA) to directly approximate the marginal posterior distribution, avoiding the issues of mixing and convergence in traditional Markov Chain Monte Carlo (MCMC) methods, and has been widely validated and applied in epidemiological research [25,26]. Additionally, to validate the stability of the forecast results, we applied the Norpred APC model [27] to predict the overall burden of CRDs and assess the changes over the next 20 years.

### Statistical analysis

All detailed statistical methods used in this study can be found in the supplementary file (S1 File). Statistical analysis and visualization were performed using R 4.4.1. We also used Joinpoint Regression Analysis software V5.2.0 and the NIH APC Web Tool for this research. The mapping was done using publicly available R packages, including ggmap and maps, based on publicly available world map data from the Natural Earth data project

(https://www.rdocumentation.org/packages/maps/versions/3.4.2). Apart from the forecasting analysis, we calculated the non-standardized rates of disease burden for individuals aged 55 and above based on GBD, reflecting the absolute burden on older populations in different regions, considering the impact of changes in age structure within the context of global aging.

## Results

### Overall burden of CRDs

In 2021, CRDs remained one of the leading causes of death globally, with this situation being more prevalent in middle-aged and older patients. Before the outbreak of the pandemic, the disease burden related to CRDs had been steadily declining worldwide [2], but this trend changed in 2021.

In 2021, we estimated that the global number of CRD cases in the population aged 55 and above reached 223 million (95% UI 206.5 to 241.5), nearly half of the total cases across all age groups [2]. The number of new cases was 18.47 million (95% UI 16.97 to 20.11) (Table 1). From a global perspective, the results of multi-model trend analysis were consistent, showing a downward trend in the global burden of CRDs between 1990 and 2021 (Table 1, Fig 1, S4 and S6 Tables in S2 File), although the burden of disease increased with age and placed more burden on males (Fig 1, S2-S4 Fig and S7 Table in S2 File). This downward trend was also observed across different regions of the world, though with some regional differences, such as a correlation between prevalence and SDI, with higher SDI regions showing increased prevalence (Fig 2). The global trend in CRD burden also differed across certain countries (Fig 3). In 11 example countries (S11-S12 and S16-S17 Tables in S2 File), the disease burden in the U.S. and the UK diverged from the global trend. In the U.S., the net drift in prevalence and incidence rates were 0.87% (95% CI 0.74% to 0.99%) and 0.85% (95% CI 0.61% to 1.1%), respectively, while the net drift in the incidence rate in the UK was 0.35% (95% CI 0.25% to 0.45%). Through APC model analysis (Fig 4, S7 Table in S2 File), we found that, within birth cohorts, older individuals born earlier were more significantly affected by environmental factors, with increased prevalence, particularly in high-SDI regions (e.g., high-income Asia-Pacific, high-income North America, Western Europe). Countries such as Japan, China, Russia, and Germany were more affected, while the U.S. showed a smaller impact (S12 and S17 Tables in S2 File). Additionally, in both global and regional local shifts (S7 Table in S2 File), the prevalence of CRDs showed a declining trend before the age of 80, with an upward trend after age 80. The stable range for these trends shifted earlier as SDI decreased. Regional differences in local shifts were observed (S12 and S17 Tables in S2 File), particularly in North America, where the U.S. exhibited an earlier shift in the stable range for CRD prevalence and incidence, with increasing trends observed across all age groups.

Affected by population aging and the COVID-19 pandemic, nearly one-third of countries and regions saw a reversal in the CRD burden trend in 2021 compared to the overall trend (S3-S5 Tables in S2 File). Pre- and post-pandemic comparisons (Fig 5, S1-S3 Table in S2 File) showed that the global EAPC for CRD prevalence rose to −0.05 (95% CI −0.19 to 0.08), with low-SDI regions experiencing more significant impacts, such as Central Asia, North Africa, the Middle East, and sub-Saharan West Africa. Meanwhile, the EAPC for global CRD incidence reversed significantly to 0.26 (95% CI 0.02 to 0.49), with middle- and high-SDI regions showing a greater impact, especially changing the incidence trend in Eastern Europe: EAPC $_{(2019–2021)}$ 1.73 (95% CI 1.07 to 2.4). Overall (S3 and S5 Tables in S2 File), during the pandemic, the absolute burden of CRDs in Asian regions, including East Asia, South Asia, and Southeast Asia, was more severe. In contrast, high-income Asia-Pacific and North American regions saw an upward trend in CRD DALYs and mortality rates due to the pandemic. Through decomposition analysis, we further found that while epidemiological changes in CRDs showed a downward trend, population growth and aging combined to increase the disease burden, especially in middle-SDI regions (S24 Table and S17 Fig in S2 File).

In the health inequality analysis, compared to 1990, the Slope Index of Inequality (SII) for CRDs decreased from 7650.18 to 4449.61, and the Lorenz curve moved closer to the diagonal, though some distance remained. This suggests

**Table 1. Comprehensive overview of the prevalence, incidence, mortality, and DALYs of CRDs at the global level and in 5 SDI regions (EAPC, Time Points: 1990, 2019, 2021).**

| Measure | Cause | Location | Cases | | | EAPC | |
|---|---|---|---|---|---|---|---|
| | | | 2019_millions(95% UI) | 2021_millions(95% UI) | % change | 2019-2021(95% CI) | 1990-2021(95% CI) |
| Prevalence | CRDs | | | | | | |
| | | Global | 211.38 (195.67 to 228.54) | 223.17 (206.50 to 241.48) | 0.06 | −0.05 (−0.19 to 0.08) | −0.71 (−0.76 to −0.65) |
| | | Low SDI | 10.36 (9.57 to 11.17) | 11.04 (10.16 to 11.94) | 0.07 | 0.45 (0.44 to 0.46) | −0.20 (−0.22 to −0.18) |
| | | Low-middle SDI | 34.66 (31.88 to 37.38) | 36.72 (33.81 to 39.61) | 0.06 | 0.23 (0.16 to 0.31) | −0.10 (−0.12 to −0.07) |
| | | Middle SDI | 56.61 (51.25 to 62.25) | 61.00 (55.26 to 67.17) | 0.08 | −0.01 (−0.14 to 0.13) | −0.40 (−0.43 to −0.37) |
| | | High-middle SDI | 44.27 (40.36 to 48.46) | 46.71 (42.61 to 51.12) | 0.06 | 0.01 (−0.02 to 0.05) | −1.05 (−1.09 to −1.00) |
| | | High SDI | 65.29 (61.70 to 69.16) | 67.51 (63.87 to 71.52) | 0.03 | 0.02 (−0.23 to 0.27) | −0.82 (−0.97 to −0.67) |
| | COPD | | | | | | |
| | | Global | 158.41 (141.70 to 175.05) | 166.94 (148.84 to 184.74) | 0.05 | −0.14 (−0.33 to 0.05) | 0.18 (0.13 to 0.23) |
| | | Low SDI | 7.06 (6.31 to 7.86) | 7.49 (6.67 to 8.35) | 0.06 | 0.19 (0.15 to 0.23) | 0.38 (0.34 to 0.42) |
| | | Low-middle SDI | 25.62 (22.82 to 28.48) | 26.98 (24.07 to 29.86) | 0.05 | −0.07 (−0.14 to −0.01) | 0.35 (0.30 to 0.40) |
| | | Middle SDI | 45.52 (39.82 to 51.32) | 48.89 (42.61 to 55.28) | 0.07 | −0.17 (−0.38 to 0.03) | 0.24 (0.19 to 0.28) |
| | | High-middle SDI | 35.45 (31.37 to 39.73) | 37.36 (32.91 to 42.05) | 0.05 | −0.04 (−0.08 to 0.00) | 0.06 (−0.01 to 0.12) |
| | | High SDI | 44.62 (41.15 to 48.16) | 46.08 (42.41 to 49.65) | 0.03 | −0.04 (−0.44 to 0.36) | 0.23 (0.17 to 0.28) |
| | Asthma | | | | | | |
| | | Global | 58.15 (51.26 to 65.28) | 61.69 (54.23 to 69.41) | 0.06 | 0.19 (0.15 to 0.22) | −2.57 (−2.76 to −2.37) |
| | | Low SDI | 3.58 (3.13 to 4.11) | 3.87 (3.37 to 4.44) | 0.08 | 1.05 (0.92 to 1.18) | −1.17 (−1.23 to −1.12) |
| | | Low-middle SDI | 10.01 (8.65 to 11.49) | 10.79 (9.35 to 12.38) | 0.08 | 1.11 (0.95 to 1.28) | −1.11 (−1.20 to −1.03) |
| | | Middle SDI | 11.94 (10.43 to 13.59) | 13.05 (11.37 to 14.88) | 0.09 | 0.73 (0.63 to 0.83) | −2.34 (−2.44 to −2.23) |
| | | High-middle SDI | 9.54 (8.48 to 10.73) | 10.10 (8.99 to 11.32) | 0.06 | 0.22 (0.20 to 0.24) | −3.79 (−3.96 to −3.63) |
| | | High SDI | 23.02 (20.57 to 25.63) | 23.81 (21.15 to 26.7) | 0.03 | 0.03 (0.00 to 0.07) | −2.40 (−2.80 to −1.99) |
| | ILD & PS | | | | | | |
| | | Global | 3.14 (2.76 to 3.58) | 3.31 (2.90 to 3.79) | 0.05 | −0.13 (−0.13 to −0.12) | 0.68 (0.58 to 0.79) |
| | | Low SDI | 0.08 (0.07 to 0.10) | 0.09 (0.08 to 0.10) | 0.12 | 0.04 (−0.28 to 0.35) | 0.20 (0.16 to 0.24) |
| | | Low-middle SDI | 0.38 (0.33 to 0.44) | 0.40 (0.34 to 0.46) | 0.05 | −0.23 (−0.57 to 0.10) | 0.49 (0.46 to 0.53) |
| | | Middle SDI | 0.60 (0.52 to 0.69) | 0.64 (0.55 to 0.74) | 0.07 | −0.29 (−0.51 to −0.08) | 0.94 (0.83 to 1.05) |
| | | High-middle SDI | 0.51 (0.45 to 0.58) | 0.54 (0.47 to 0.61) | 0.06 | 0.02 (−0.22 to 0.25) | 0.92 (0.76 to 1.09) |
| | | High SDI | 1.57 (1.38 to 1.77) | 1.64 (1.44 to 1.87) | 0.04 | 0.71 (0.47 to 0.96) | 0.95 (0.84 to 1.06) |
| Incidence | CRDs | | | | | | |
| | | Global | 17.39 (15.95 to 18.93) | 18.47 (16.97 to 20.11) | 0.06 | 0.26 (0.02 to 0.49) | −0.59 (−0.64 to −0.54) |
| | | Low SDI | 0.93 (0.85 to 1.03) | 0.99 (0.90 to 1.09) | 0.06 | 0.33 (0.16 to 0.50) | −0.21 (−0.25 to −0.18) |
| | | Low-middle SDI | 3.21 (2.95 to 3.49) | 3.40 (3.12 to 3.69) | 0.06 | 0.15 (0.14 to 0.15) | −0.27 (−0.31 to −0.24) |
| | | Middle SDI | 5.19 (4.72 to 5.70) | 5.60 (5.11 to 6.15) | 0.08 | 0.13 (0.12 to 0.13) | −0.67 (−0.72 to −0.63) |
| | | High-middle SDI | 3.54 (3.23 to 3.86) | 3.76 (3.43 to 4.11) | 0.06 | 0.29 (0.19 to 0.39) | −1.00 (−1.06 to −0.93) |
| | | High SDI | 4.50 (4.12 to 4.88) | 4.70 (4.31 to 5.12) | 0.04 | 0.59 (−0.33 to 1.50) | −0.43 (−0.58 to −0.28) |
| | COPD | | | | | | |
| | | Global | 12.35 (11.15 to 13.47) | 13.07 (11.76 to 14.25) | 0.06 | 0.09 (−0.04 to 0.21) | 0.07 (0.03 to 0.11) |
| | | Low SDI | 0.57 (0.52 to 0.63) | 0.61 (0.55 to 0.66) | 0.07 | 0.13 (0.12 to 0.14) | 0.23 (0.20 to 0.26) |
| | | Low-middle SDI | 2.16 (1.96 to 2.34) | 2.28 (2.06 to 2.47) | 0.06 | −0.04 (−0.15 to 0.08) | 0.14 (0.09 to 0.19) |
| | | Middle SDI | 3.81 (3.39 to 4.20) | 4.11 (3.65 to 4.53) | 0.08 | −0.01 (−0.14 to 0.11) | −0.05 (−0.10 to −0.01) |
| | | High-middle SDI | 2.74 (2.43 to 3.03) | 2.90 (2.56 to 3.21) | 0.06 | 0.23 (0.22 to 0.24) | −0.15 (−0.21 to −0.09) |
| | | High SDI | 3.06 (2.80 to 3.32) | 3.17 (2.90 to 3.47) | 0.04 | 0.21 (−0.07 to 0.50) | 0.32 (0.28 to 0.35) |

*(Continued)*

**Table 1.** (Continued)

| Measure | Cause | Location | Cases | | | EAPC | |
|---|---|---|---|---|---|---|---|
| | | | 2019_millions(95% UI) | 2021_millions(95% UI) | % change | 2019-2021(95% CI) | 1990-2021(95% CI) |
| | Asthma | | | | | | |
| | | Global | 4.72 (3.88 to 5.71) | 5.06 (4.15 to 6.12) | 0.07 | 0.73 (−0.49 to 1.96) | −1.96 (−2.11 to −1.81) |
| | | Low SDI | 0.35 (0.28 to 0.43) | 0.37 (0.30 to 0.46) | 0.06 | 0.67 (0.22 to 1.11) | −0.85 (−0.90 to −0.80) |
| | | Low-middle SDI | 1.00 (0.82 to 1.22) | 1.07 (0.87 to 1.32) | 0.07 | 0.56 (0.34 to 0.78) | −1.04 (−1.11 to −0.98) |
| | | Middle SDI | 1.30 (1.07 to 1.56) | 1.42 (1.17 to 1.70) | 0.09 | 0.56 (0.21 to 0.90) | −2.09 (−2.19 to −2.00) |
| | | High-middle SDI | 0.75 (0.63 to 0.90) | 0.80 (0.67 to 0.96) | 0.07 | 0.51 (0.08 to 0.94) | −3.13 (−3.29 to −2.96) |
| | | High SDI | 1.31 (1.05 to 1.63) | 1.39 (1.12 to 1.73) | 0.06 | 1.46 (−2.37 to 5.44) | −1.84 (−2.27 to −1.4) |
| | ILD & PS | | | | | | |
| | | Global | 0.27 (0.23 to 0.31) | 0.28 (0.24 to 0.33) | 0.04 | −0.27 (−0.50 to −0.04) | 1.21 (1.07 to 1.35) |
| | | Low SDI | 0.01 (0.01 to 0.01) | 0.01 (0.01 to 0.01) | 0.00 | −0.08 (−0.09 to −0.07) | 0.32 (0.28 to 0.35) |
| | | Low-middle SDI | 0.04 (0.04 to 0.05) | 0.05 (0.04 to 0.06) | 0.25 | −0.36 (−0.48 to −0.23) | 0.54 (0.51 to 0.57) |
| | | Middle SDI | 0.06 (0.05 to 0.07) | 0.06 (0.05 to 0.07) | 0.00 | −0.40 (−0.44 to −0.37) | 1.46 (1.33 to 1.59) |
| | | High-middle SDI | 0.04 (0.03 to 0.04) | 0.04 (0.03 to 0.05) | 0.00 | 0.05 (−0.18 to 0.29) | 1.72 (1.51 to 1.93) |
| | | High SDI | 0.12 (0.11 to 0.14) | 0.13 (0.11 to 0.15) | 0.08 | 0.44 (−0.07 to 0.94) | 1.56 (1.38 to 1.74) |
| Deaths | CRDs | | | | | | |
| | | Global | 4.03 (3.68 to 4.35) | 4.15 (3.76 to 4.58) | 0.03 | −1.29 (−1.57 to −1.01) | −1.40 (−1.50 to −1.30) |
| | | Low SDI | 0.32 (0.28 to 0.36) | 0.32 (0.28 to 0.37) | 0.00 | −2.96 (−3.32 to −2.59) | −0.29 (−0.39 to −0.18) |
| | | Low-middle SDI | 1.16 (1.03 to 1.29) | 1.16 (1.03 to 1.30) | 0.00 | −2.68 (−2.85 to −2.51) | −0.11 (−0.19 to −0.02) |
| | | Middle SDI | 1.31 (1.17 to 1.46) | 1.39 (1.21 to 1.57) | 0.06 | −0.93 (−1.35 to −0.51) | −2.43 (−2.57 to −2.29) |
| | | High-middle SDI | 0.69 (0.60 to 0.77) | 0.72 (0.62 to 0.82) | 0.04 | −0.36 (−0.45 to −0.28) | −2.65 (−2.90 to −2.40) |
| | | High SDI | 0.55 (0.48 to 0.59) | 0.56 (0.49 to 0.60) | 0.02 | −0.32 (−1.33 to 0.70) | −0.55 (−0.64 to −0.46) |
| | COPD | | | | | | |
| | | Global | 3.46 (3.15 to 3.73) | 3.59 (3.22 to 3.94) | 0.04 | −1.04 (−1.32 to −0.77) | −1.46 (−1.57 to −1.35) |
| | | Low SDI | 0.25 (0.22 to 0.28) | 0.25 (0.22 to 0.28) | 0.00 | −2.76 (−3.21 to −2.32) | −0.08 (−0.19 to 0.03) |
| | | Low-middle SDI | 0.95 (0.85 to 1.04) | 0.95 (0.85 to 1.05) | 0.00 | −2.43 (−2.58 to −2.27) | 0.05 (−0.04 to 0.13) |
| | | Middle SDI | 1.18 (1.05 to 1.33) | 1.25 (1.08 to 1.43) | 0.06 | −0.75 (−1.19 to −0.32) | −2.51 (−2.65 to −2.36) |
| | | High-middle SDI | 0.64 (0.56 to 0.72) | 0.67 (0.58 to 0.76) | 0.05 | −0.23 (−0.28 to −0.18) | −2.68 (−2.94 to −2.41) |
| | | High SDI | 0.45 (0.39 to 0.48) | 0.46 (0.40 to 0.49) | 0.02 | −0.23 (−1.22 to 0.77) | −0.58 (−0.67 to −0.48) |
| | Asthma | | | | | | |
| | | Global | 0.36 (0.29 to 0.47) | 0.35 (0.28 to 0.45) | −0.03 | −3.33 (−3.58 to −3.08) | −1.80 (−1.88 to −1.72) |
| | | Low SDI | 0.06 (0.04 to 0.09) | 0.05 (0.04 to 0.08) | −0.17 | −3.98 (−4.01 to −3.96) | −1.20 (−1.31 to −1.09) |
| | | Low-middle SDI | 0.17 (0.12 to 0.26) | 0.17 (0.12 to 0.24) | 0.00 | −4.21 (−4.47 to −3.95) | −1.03 (−1.13 to −0.93) |
| | | Middle SDI | 0.09 (0.08 to 0.10) | 0.09 (0.08 to 0.10) | 0.00 | −2.65 (−2.96 to −2.34) | −2.43 (−2.60 to −2.25) |
| | | High-middle SDI | 0.02 (0.02 to 0.02) | 0.02 (0.02 to 0.02) | 0.00 | −1.79 (−1.85 to −1.73) | −3.99 (−4.22 to −3.76) |
| | | High SDI | 0.01 (0.01 to 0.02) | 0.01 (0.01 to 0.02) | 0.00 | −0.44 (−1.92 to 1.07) | −5.51 (−5.96 to −5.05) |
| | ILD & PS | | | | | | |
| | | Global | 0.17 (0.15 to 0.19) | 0.17 (0.15 to 0.20) | 0.00 | −1.87 (−2.52 to −1.22) | 1.89 (1.72 to 2.05) |
| | | Low SDI | 0.01 (0.01 to 0.01) | 0.01 (0.01 to 0.01) | 0.00 | −2.19 (−2.59 to −1.79) | 0.74 (0.62 to 0.86) |
| | | Low-middle SDI | 0.03 (0.02 to 0.05) | 0.03 (0.02 to 0.05) | 0.00 | −2.03 (−2.12 to −1.94) | 1.11 (1.01 to 1.21) |
| | | Middle SDI | 0.03 (0.02 to 0.03) | 0.03 (0.02 to 0.04) | 0.00 | −2.27 (−2.63 to −1.91) | 1.59 (1.47 to 1.71) |
| | | High-middle SDI | 0.02 (0.02 to 0.02) | 0.02 (0.02 to 0.02) | 0.00 | −2.72 (−3.27 to −2.17) | 1.77 (1.60 to 1.95) |
| | | High SDI | 0.08 (0.07 to 0.08) | 0.08 (0.07 to 0.09) | 0.00 | −0.56 (−2.06 to 0.95) | 2.88 (2.62 to 3.14) |
| DALYs | CRDs | | | | | | |
| | | Global | 81.19 (76.08 to 86.79) | 83.67 (77.49 to 90.36) | 0.03 | −1.25 (−1.59 to −0.92) | −1.59 (−1.67 to −1.50) |
| | | Low SDI | 6.93 (6.19 to 7.80) | 6.97 (6.19 to 7.95) | 0.01 | −2.41 (−2.46 to −2.36) | −0.58 (−0.65 to −0.51) |
| | | Low-middle SDI | 24.19 (21.82 to 26.54) | 24.24 (21.68 to 26.79) | 0.00 | −2.52 (−2.83 to −2.21) | −0.43 (−0.49 to −0.37) |

*(Continued)*

**Table 1.** (Continued)

| Measure | Cause | Location | Cases | | | EAPC | |
|---|---|---|---|---|---|---|---|
| | | | 2019_millions(95% UI) | 2021_millions(95% UI) | % change | 2019-2021(95% CI) | 1990-2021(95% CI) |
| | | Middle SDI | 25.33 (23.04 to 28.04) | 26.79 (24.05 to 29.87) | 0.06 | −0.95 (−1.34 to −0.55) | −2.60 (−2.73 to −2.48) |
| | | High-middle SDI | 12.90 (11.59 to 14.20) | 13.52 (12.15 to 15.03) | 0.05 | −0.34 (−0.47 to −0.21) | −2.87 (−3.08 to −2.65) |
| | | High SDI | 11.79 (10.86 to 12.57) | 12.11 (11.15 to 12.93) | 0.03 | −0.32 (−1.06 to 0.43) | −0.84 (−0.92 to −0.76) |
| | COPD | | | | | | |
| | | Global | 67.93 (63.15 to 72.58) | 70.28 (64.85 to 76.00) | 0.03 | −1.06 (−1.41 to −0.71) | −1.60 (−1.70 to −1.50) |
| | | Low SDI | 5.34 (4.81 to 5.89) | 5.39 (4.87 to 6.00) | 0.01 | −2.25 (−2.35 to −2.15) | −0.36 (−0.43 to −0.29) |
| | | Low-middle SDI | 19.38 (17.75 to 21.18) | 19.51 (17.77 to 21.32) | 0.01 | −2.31 (−2.61 to −2.00) | −0.26 (−0.33 to −0.20) |
| | | Middle SDI | 22.31 (20.13 to 24.79) | 23.64 (21.06 to 26.47) | 0.06 | −0.83 (−1.24 to −0.42) | −2.68 (−2.81 to −2.55) |
| | | High-middle SDI | 11.64 (10.43 to 12.90) | 12.22 (10.94 to 13.70) | 0.05 | −0.24 (−0.41 to −0.07) | −2.85 (−3.07 to −2.62) |
| | | High SDI | 9.22 (8.53 to 9.78) | 9.48 (8.71 to 10.05) | 0.03 | −0.25 (−1.04 to 0.54) | −0.65 (−0.72 to −0.58) |
| | Asthma | | | | | | |
| | | Global | 9.04 (7.34 to 11.6) | 9.08 (7.45 to 11.28) | 0.00 | −2.52 (−2.83 to −2.2) | −2.19 (−2.26 to −2.13) |
| | | Low SDI | 1.30 (0.93 to 1.91) | 1.28 (0.94 to 1.86) | −0.02 | −3.21 (−3.36 to −3.05) | −1.49 (−1.58 to −1.41) |
| | | Low-middle SDI | 3.83 (2.83 to 5.60) | 3.74 (2.81 to 5.29) | −0.02 | −3.74 (−4.13 to −3.36) | −1.35 (−1.41 to −1.28) |
| | | Middle SDI | 2.15 (1.90 to 2.44) | 2.23 (1.97 to 2.56) | 0.04 | −1.86 (−2.20 to −1.51) | −2.59 (−2.73 to −2.45) |
| | | High-middle SDI | 0.70 (0.58 to 0.87) | 0.73 (0.60 to 0.89) | 0.04 | −0.76 (−0.96 to −0.55) | −4.23 (−4.44 to −4.03) |
| | | High SDI | 1.07 (0.78 to 1.41) | 1.10 (0.79 to 1.45) | 0.03 | −0.22 (−0.56 to 0.13) | −3.51 (−3.90 to −3.12) |
| | ILD & PS | | | | | | |
| | | Global | 3.21 (2.78 to 3.62) | 3.28 (2.85 to 3.66) | 0.02 | −1.61 (−2.26 to −0.96) | 1.26 (1.13 to 1.39) |
| | | Low SDI | 0.20 (0.12 to 0.28) | 0.20 (0.12 to 0.28) | 0.00 | −1.88 (−2.02 to −1.73) | 0.44 (0.36 to 0.52) |
| | | Low-middle SDI | 0.71 (0.48 to 0.97) | 0.72 (0.49 to 0.98) | 0.01 | −1.95 (−2.19 to −1.72) | 0.77 (0.70 to 0.84) |
| | | Middle SDI | 0.58 (0.50 to 0.70) | 0.61 (0.52 to 0.74) | 0.05 | −1.43 (−1.67 to −1.18) | 1.15 (1.07 to 1.23) |
| | | High-middle SDI | 0.40 (0.36 to 0.44) | 0.40 (0.36 to 0.45) | 0.00 | −2.12 (−2.32 to −1.93) | 0.97 (0.85 to 1.09) |
| | | High SDI | 1.32 (1.19 to 1.41) | 1.35 (1.21 to 1.45) | 0.02 | −0.56 (−1.91 to 0.81) | 2.06 (1.84 to 2.28) |

EAPC = Estimated Annual Percentage Change; CRDs = Chronic Respiratory Diseases; COPD = Chronic Obstructive Pulmonary Disease; ILD & PS = Interstitial lung disease and pulmonary sarcoidosis; DALYs = Disability-Adjusted Life Years; SDI = Sociodemographic Index

that while health inequality has improved, it still exists, particularly in higher-SDI countries (Fig 6). In the forecast models, two prediction methods were used to predict the changes in CRDs over the next 10 and 20 years. The results of the two models were almost identical, suggesting that while the global incidence and prevalence of CRDs may continue to decline, the absolute burden will still increase slowly (Fig 7-8, S21 Table in S2 File).

## Overall burden of COPD

In 2021, we estimated that the global absolute number of COPD cases in individuals aged 55 and above would reach 166 million (95% UI 148.84 to 184.74), with 13.07 million new cases (95% UI 11.76 to 14.25), 3.59 million deaths, and 70.28 million years of life lost. Compared to the overall burden of COPD in the global population in 2019, the absolute burden in middle-aged and older adults accounted for nearly 80% of the total [2](Table 1). Unlike the overall trend for CRDs, globally, the prevalence and incidence of COPD are on the rise, with the AAPC for prevalence at 0.18% (95% CI 0.16% to 0.20%) and for incidence at 0.11% (95% CI 0.09% to 0.14%). However, DALYs and mortality rates primarily show a downward trend (Table 1, Fig 1, S4 Table in S2 File). This trend is also observed across different regions globally (Fig 5, S3–S5 and S8 Tables in S2 File), with the highest prevalence in high-SDI regions, especially in high-income North America (the U.S.), while the highest incidence is found in mid- and low-SDI regions, primarily concentrated in South Asia (India). Additionally, the trend shows an accelerated increase in prevalence in low-SDI regions (e.g., Lebanon in North

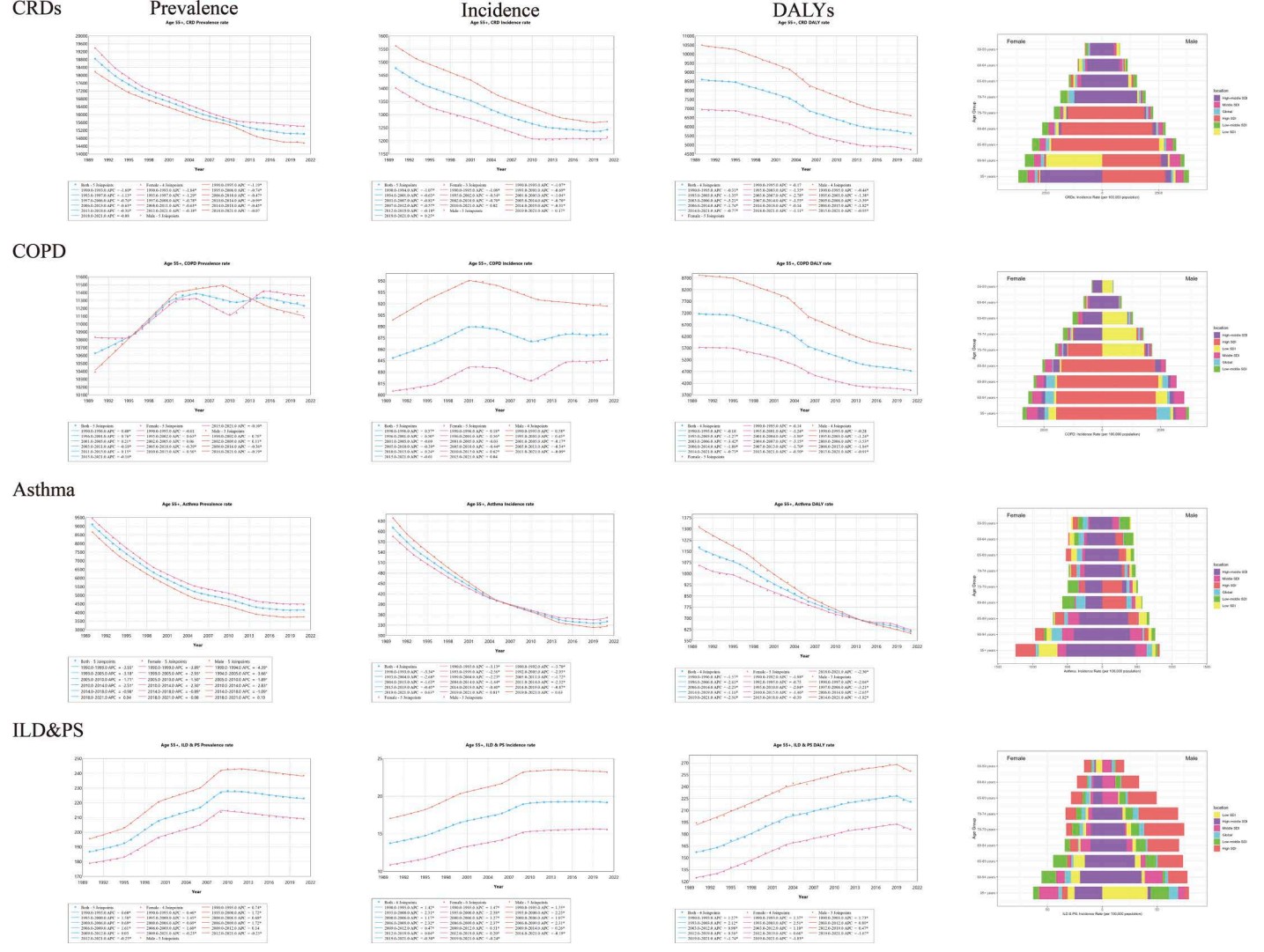

**Fig 1. Joinpoint regression analysis figures for global prevalence, incidence, and DALYs of CRDs, COPD, Asthma, and ILD & PS, along with Age-Gender Pyramids for Their Incidence.**

Africa and the Middle East) and an accelerated increase in incidence in high-SDI regions (e.g., parts of Europe and the Middle East, including Russia and Saudi Arabia). The APC model's analysis results are similar to the changes seen in CRDs (S11,S13,S16 and S18 Tables in S2 File), but there are differences. As with CRDs, the burden of COPD is heavier among males, but the period effect analysis reveals that the burden of COPD in females is gradually increasing, whereas it is decreasing in males. Compared to CRDs, the birth cohort of COPD shows a significant impact in high-SDI regions like Japan and Russia, with birth period being a key factor, while more recent generations in low-SDI regions are experiencing increased risks of COPD. Local shift results further indicate that, compared to CRDs, the overall age of onset for COPD has shifted earlier (to ages 70–74). In 11 example countries, the U.S., Germany, the UK, and Saudi Arabia have poorly controlled COPD prevalence and incidence, with varying degrees of upward trends in local shifts across all age groups. In contrast, Japan and China have managed the COPD burden well.

**Fig 2. Trends in the prevalence burden of CRDs with changes in SDI at the global level, 5 SDI regions, and 21 GBD regions.**

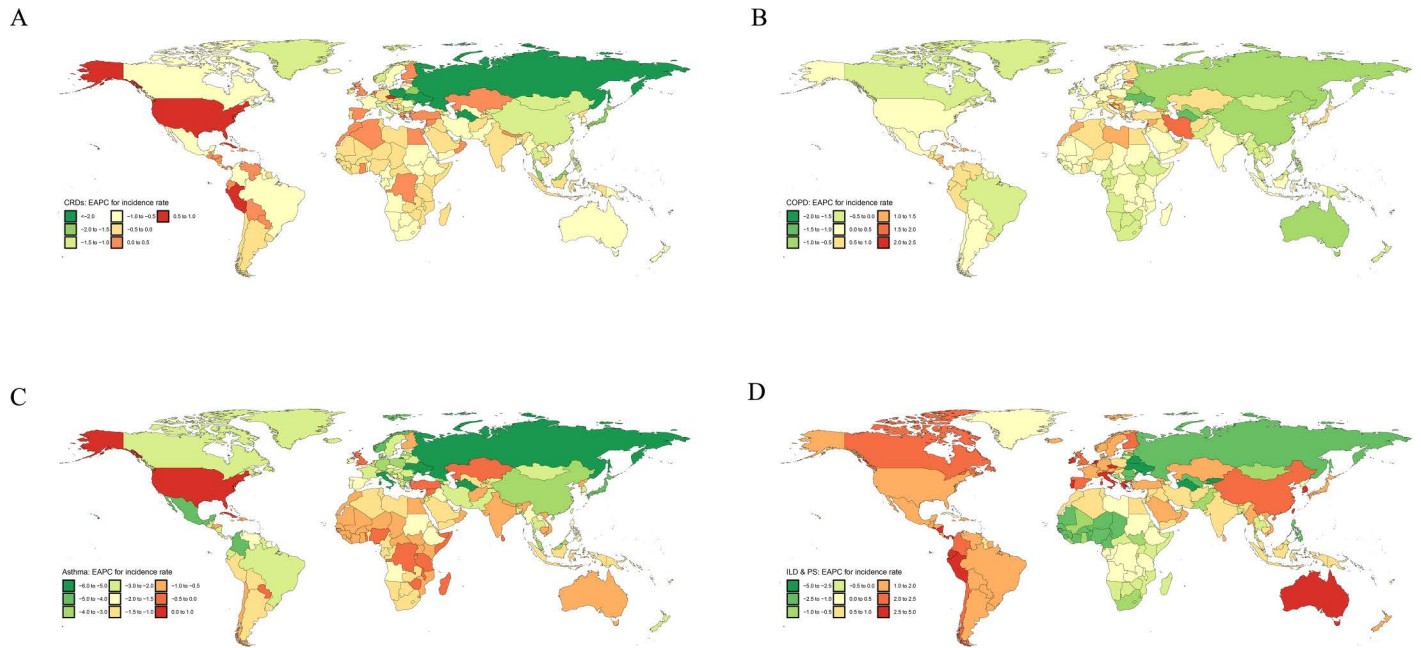

**Fig 3. Trends in the incidence of CRDs, COPD, asthma, and ILD & PS over the past 30 Years in 204 countries globally; EAPC(1990-2021); A: CRDs EAPC for incidence rate; B: COPD EAPC for incidence rate; C: Asthma EAPC for incidence rate; D: ILD & PS EAPC for incidence rate; The mapping was done using publicly available R packages, based on publicly available world map data from the Natural Earth data project (https://www.rdocumentation.org/packages/maps/versions/3.4.2).**

During the pandemic, the EAPC for COPD prevalence declined to −0.14 (95% CI −0.33 to 0.05) compared to the overall trend, but this was not statistically significant. The EAPC for other burden types showed a slight increase, but it was not substantial (Table 1). There were, of course, regional differences; in 2019–2021 (S3 and S5 Tables in S2 File), middle- and high-SDI regions were most affected, with a reversal in the incidence trend, showing an EAPC of 0.23 (95% CI 0.22 to 0.24), mainly reflecting a faster increase in prevalence and incidence in Eastern Europe (Russia). At the same time, changes in disability and mortality risks were also more pronounced in middle- and high-SDI regions, with higher mortality and DALY burdens observed in high-income Asia-Pacific regions. Through

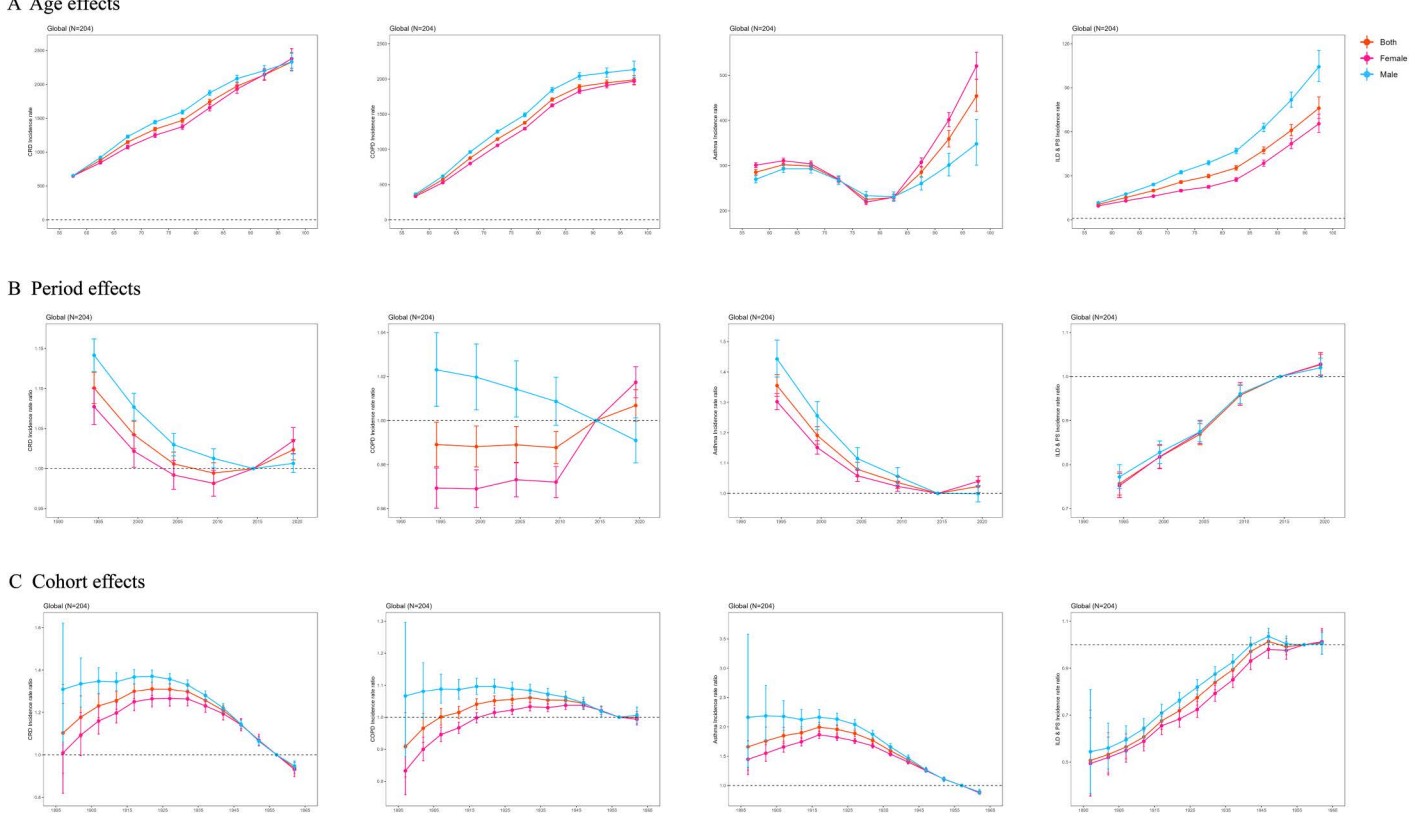

**Fig 4. APC model analysis figures for the incidence of CRDs, COPD, Asthma, and ILD & PS at the global level; A: Age effect figures for the incidence of CRDs, COPD, Asthma, and ILD & PS at the global level; B: Period effect figures for the incidence of CRDs, COPD, Asthma, and ILD & PS at the global level; C: Cohort effect figures for the incidence of CRDs, COPD, Asthma, and ILD & PS at the global level.**

decomposition analysis, we found that, compared to CRDs, COPD is more significantly influenced by population aging, contributing to higher incidence, prevalence, and disability-mortality risks in middle- and high-SDI regions (S25 Table and S17 Fig in S2 File).

In the health inequality analysis (Fig 6), compared to 1990, the Slope Index increased from 3567.00 to 3609.64, and the Lorenz curve moved further away from the diagonal. This indicates that health inequality continues to exist and gradually widens in higher-SDI countries. The BAPC forecast model (Fig 8, S21 Table in S2 File) suggests that within ten years, the prevalence of COPD will gradually decline, but the incidence will see a small increase.

## Overall Burden of Asthma

In 2021, the global number of asthma cases in individuals aged 55 and above is expected to reach 61.69 million (95% UI 54.23 to 69.41), with 5.06 million new cases (95% UI 4.15 to 6.12), leading to 350,000 deaths and 9.08 million years of life lost (Table 1). The burden of asthma in terms of prevalence and incidence primarily affects middle- and high-SDI regions, especially in Asia (particularly South Asia) and high-income North America, such as the U.S. and India (S3 and S5 Tables in S2 File). However, in middle- and low-SDI regions, asthma has caused more disability and mortality risks (Table 1). Overall, the global burden of asthma has shown a downward trend over the past 30 years (Fig 1), and this trend remains consistent across most countries and regions. Unlike the overall situation for CRDs and COPD, the burden of asthma is

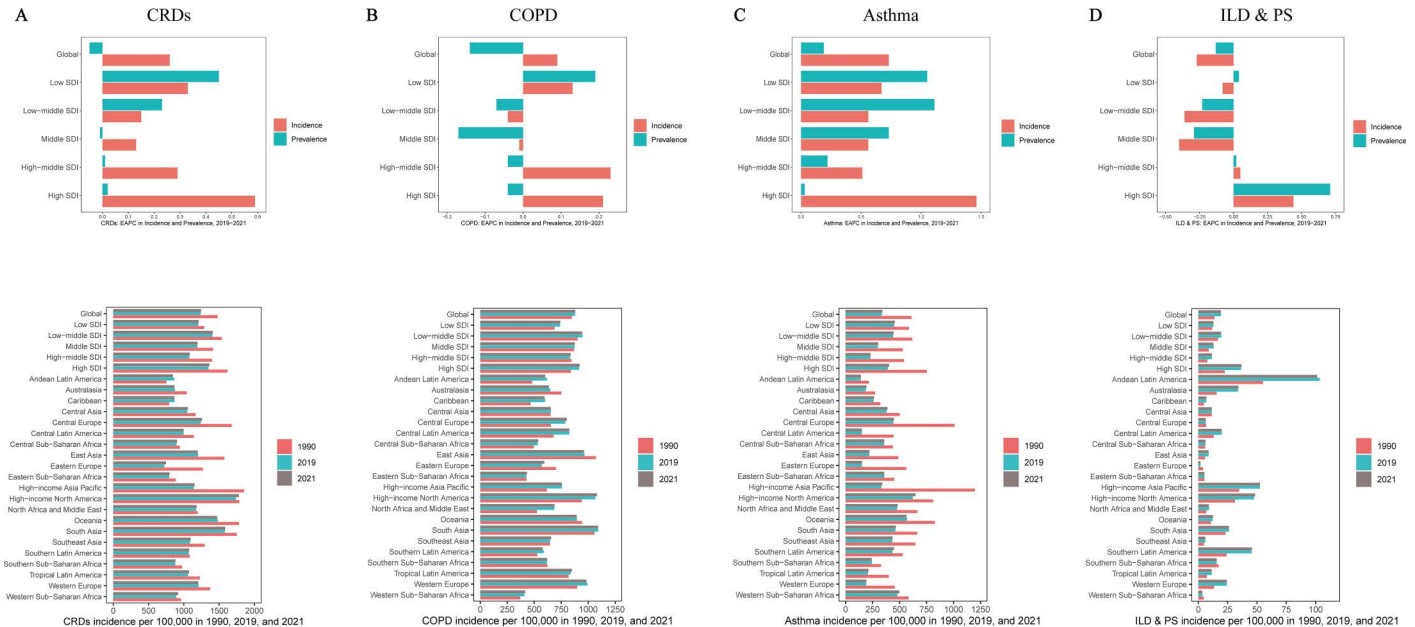

**Fig 5. Prevalence and incidence of CRDs, COPD, Asthma, and ILD & PS at the Global Level, 5 SDI regions, and 21 GBD regions; EAPC: 2019-2021.**

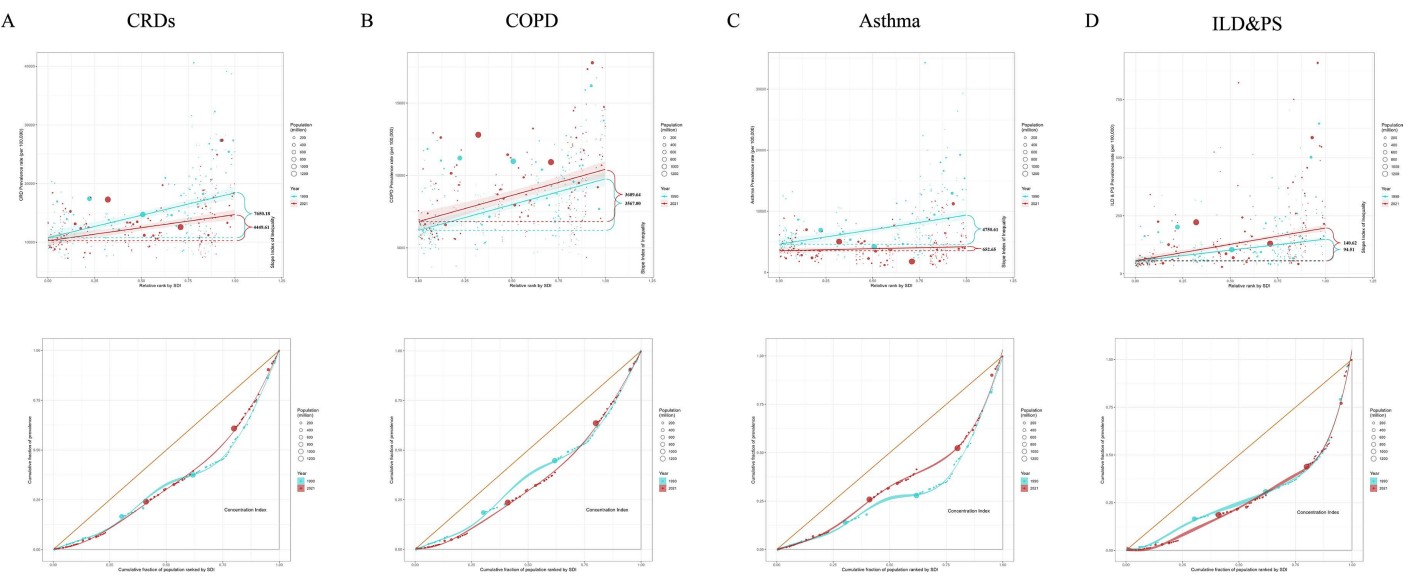

**Fig 6. Global cross-country analysis of health inequalities in the prevalence of CRDs, COPD, Asthma, and ILD & PS.**

heavier in elderly women (Fig 1, S4 Table in S2 File). Although the burden of asthma continues to decline in most regions of the world, some differences exist. In the U.S., the absolute burden of asthma not only remains the highest but also increases the fastest, with a net drift of 1.95%. The UK also shows an upward trend in asthma incidence, with a net drift of 0.98% (S5, S11-S16 Tables in S2 File). In the APC model analysis (S14 and S19 Tables in S2 File), asthma shows a

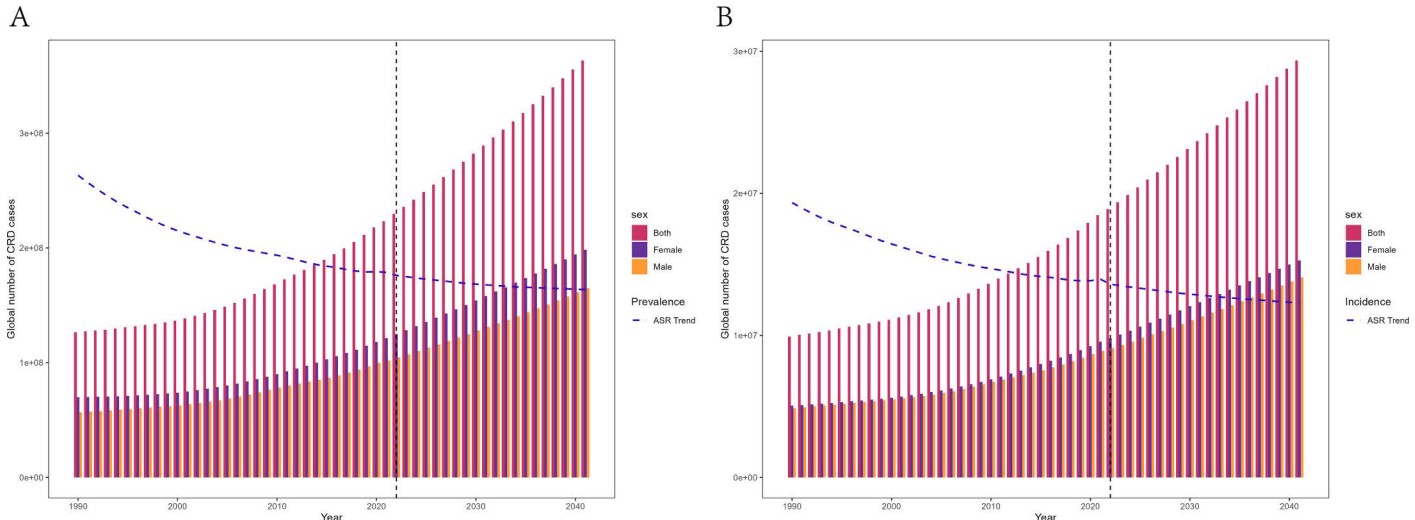

**Fig 7. Projected trends in the global prevalence and incidence burden of CRDs Over the next 20 years in terms of cases and rate.**

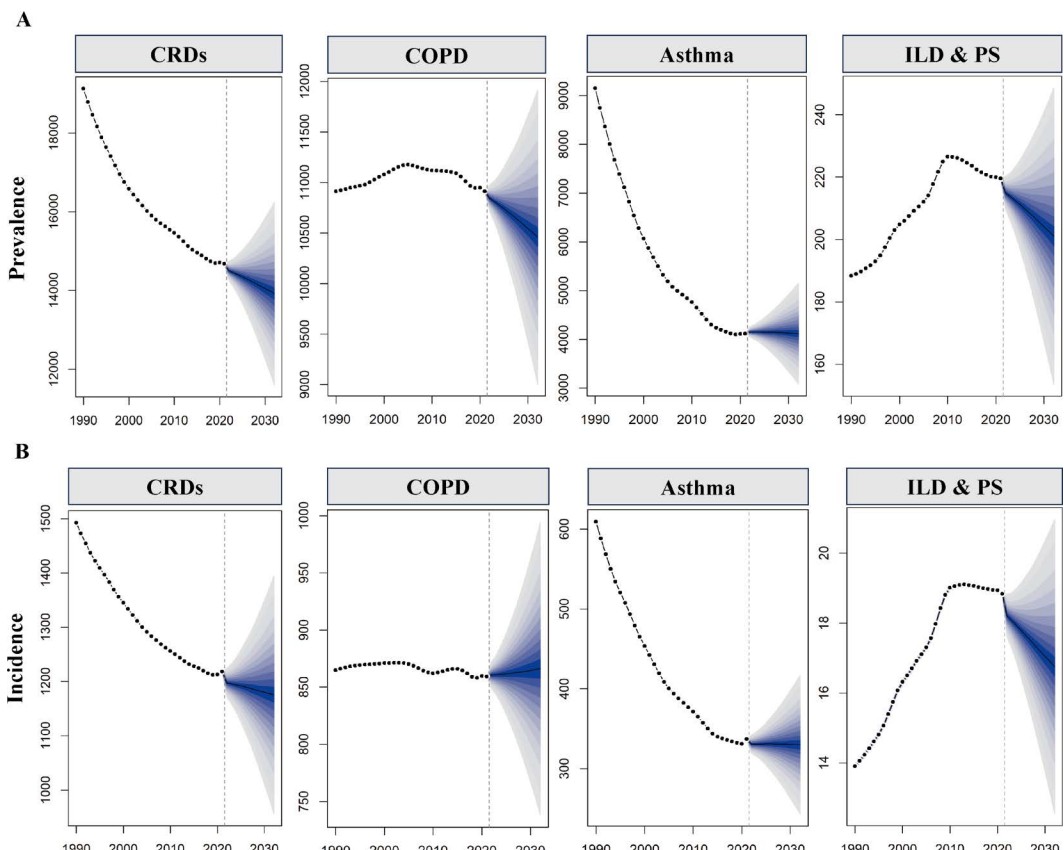

**Fig 8. BAPC projected trends in the prevalence and incidence burden of CRDs, COPD, Asthma, and ILD & PS over the next 10 years at the global level.**

similar trend to CRDs overall, with individuals born earlier having a higher risk, primarily affecting middle- and high-SDI countries such as Germany, Japan, Russia, and China, while the U.S. is not affected by the birth period. Unlike CRDs and COPD, decomposition analysis (S17 Fig in S2 File) and local shift results suggest that the risk of asthma in terms of prevalence and incidence does not increase in older age groups, but does lead to higher disability and mortality risks.

Asthma is the major contributor to the global decline in CRD's burden. In 2019−2021 (Table 1), the prevalence and incidence of asthma in more than half of the regions experienced a significant reversal, with global EAPC for prevalence and incidence reaching 0.19 (95% CI 0.15 to 0.22) and 0.73 (95% CI −0.49 to 1.96), respectively. However, the disability and mortality caused by asthma have been decreasing. Similar to the global trend in CRDs (S3 Table in S2 File), during the pandemic, the incidence of asthma increased the fastest in high-SDI regions, with North America's high-income region seeing the highest increase in incidence at 2.13 (EAPC). Conversely, the prevalence increased the fastest in middle- and low-SDI regions, with sub-Saharan West Africa showing the highest increase in prevalence at 1.81 (EAPC). In Asia, asthma caused more disability and death, with the high-income Asia-Pacific region showing the highest increase in mortality and DALYs, at 3.38 (EAPC) and 1.29 (EAPC), respectively. Decomposition analysis (S26 Table and S17 Fig in S2 File) suggests that population aging is not a driving factor for asthma prevalence and incidence, but it has a significant impact on disability and mortality in elderly individuals, especially in middle- and low-SDI regions.

In the health inequality analysis (Fig 6), compared to 1990, the slope index decreased from 4750.61 to 652.65, and the Lorenz curve moved closer to the diagonal. This indicates that health inequality in asthma has significantly decreased over the past 30 years globally. The forecasting model suggests that, over the next 10 years, the prevalence and incidence of asthma will remain stable with a slight decrease (Fig 8, S21 Table in S2 File).

**Overall burden of ILD&PS**

In 2021 (Table 1), the global number of ILD & PS cases in individuals aged 55 and above is expected to reach 3.31 million (95% UI 2.89 to 3.79), with 280,000 new cases (95% UI 244,800–328,800), resulting in 170,000 deaths and 3.28 million years of life lost. Among the global population, nearly three-quarters of ILD & PS cases are found in individuals aged 55 and above [2]. The burden of ILD & PS varies across different regions. In high-SDI regions, the overall burden rate for ILD & PS is higher, with Japan having a prevalence rate of 908 per 100,000 people. In middle- and low-SDI regions, the absolute burden of ILD & PS is heavier (Fig 5, S3 Table and S1 Fig in S2 File), particularly in South Asia (S3 Table in S2 File), where ILD & PS causes more incidence, disability, and mortality. However, there is an exception in the high-income U.S., where the number of cases is the highest, reaching 587,000 (S5 Table in S2 File). Similar to COPD, the global burden of ILD & PS has shown an upward trend from 1990 to 2021, with a greater impact on elderly men. The mortality rate increased the fastest, with an AAPC of 1.65% (95% CI 1.47% to 1.84%), followed by the DALY rate at 1.10% (95% CI 0.97% to 1.23%). All SDI regions follow the global trend, with the highest increases in prevalence and incidence observed in high-SDI regions such as Andean Latin America and Australasia, and the fastest increases in mortality and DALYs in Western Europe and Australasia (S3 Table in S2 File). Unlike other CRDs, the birth cohort analysis (S9-S10 Table in S2 File) for ILD & PS shows that people born in recent years are at higher risk of ILD & PS across the globe. However, in middle- and low-SDI regions, including sub-Saharan West Africa, tropical Latin America, and Eastern Europe, the birth cohort analysis indicates that those born earlier are more likely to develop the disease, especially in Eastern Europe. Local shift results indicate that in most countries and regions worldwide, the prevalence and incidence of ILD & PS are increasing across all age groups in individuals aged 55 and above, with a significant impact in middle- and high-income regions like the UK and China (S10, S15 and S20 Tables in S2 File). However, in some parts of Eastern Europe and sub-Saharan Africa, there is no shift in the trend for ILD & PS. APC model analysis suggests that the unique trend observed in Eastern Europe is largely driven by Russia. In Russia, unlike other regions, individuals

born earlier were more affected by the environment at the time, while the burden in those born in recent years is steadily decreasing (S15 and S20 Tables in S2 File).

From 2019 to 2021, compared to the overall trend, the burden of ILD & PS globally decreased, with the fastest decline in the mortality rate, which had an AAPC of −2.02% (95% CI −3.64 to −0.38), followed by DALYs at −1.74% (95% CI −2.95% to −0.53%) (Table 1, S4 Table in S2 File). Although the burden of ILD & PS has decreased in many regions, the incidence and prevalence have risen the fastest in high-income North America, with EAPC values of 1.63 and 1.06, respectively. Central Europe showed the highest increase in mortality (0.90 EAPC), while sub-Saharan Southern Africa had the highest increase in DALYs (0.89 EAPC) (S3 Table in S2 File). Decomposition analysis suggests that globally, the epidemiological trends for all burdens of ILD & PS are still increasing, with population aging also having a significant impact on middle- and high-SDI regions (S27 Table and S17 Fig in S2 File).

In the health inequality analysis (Fig 6), compared to 1990, the slope index increased from 94.51 to 140.62. The Lorenz curve also deviated slightly, with a significant distance from the diagonal. This indicates that health inequality remains and is widening. The forecasting model suggests that over the next ten years, the global burden of ILD & PS is unlikely to increase and may show a downward trend (Fig 8, S21 Table in S2 File).

### Risk factor assessment

We conducted an analysis of the risk factors associated with asthma, COPD, and the overall burden of CRDs, including their impact on disability and mortality risks.

In the global assessment of CRD risk factors (S22-S23 Tables in S2 File), air pollution had the greatest impact on DALYs and mortality rates, with a contribution of 2225.06 years per 100,000 person-years (95% UI 1799.22 to 2664.99) and 115.22 deaths per 100,000 people (95% UI 92.01 to 140.25), with particulate matter pollution being the most significant. Smoking behavior followed, with notable gender differences. However, as SDI increases, the risk from smoking gradually rises and surpasses the harm caused by air pollution. Tobacco is a major risk factor in Europe, high-income Asia-Pacific, and North America, with smoking being the primary risk factor for men. Environmental particulate pollution affects women more severely than men and women are more susceptible to household air pollution. Temperature has a lower impact on CRDs, but low temperatures are more likely to harm elderly populations. The risk factors for CRDs differ across regions, with many low-income countries experiencing more severe impacts. In the U.S., Japan, and Cuba, the impact of tobacco is significant. In China, besides tobacco, environmental particulate matter also plays a considerable role. In North Korea and India, the harm caused by air pollution is greater than that caused by tobacco (Fig 9, S5-S6 and S5-S12 Figs in S2 File).

COPD is influenced by risk factors in a similar way to CRDs overall [2], with air pollution having a more significant impact on middle- and low-SDI regions, especially affecting women more. Tobacco, as the second most significant risk factor, primarily affects high-SDI regions, including all of Europe, high-income Asia-Pacific, and North America, as well as parts of Latin America. For further details, refer to the supplementary materials (Fig 9, S7-S8 and S13-S14 Figs in S2 File).

Asthma is influenced by risk factors quite differently from COPD, with metabolic factors—particularly high body mass index (BMI)—having a significant impact on middle-aged and older individuals globally, with a heavier burden on women with a high BMI. However, smoking remains the largest risk factor among elderly men. Similar to other CRDs, environmental pollution mainly increases the burden of asthma in low-SDI regions such as sub-Saharan Africa, while smoking primarily increases the risk in middle- and high-SDI regions such as Asia. For further details, refer to the supplementary materials (Fig 9, S9-S10 and S15-S16 Figs in S2 File).

### Discussion

Through the construction of multiple models, this study evaluates the changes in the comprehensive burden of CRDs among individuals aged 55 and above at the global, regional, and national levels. It also conducts a cross-regional

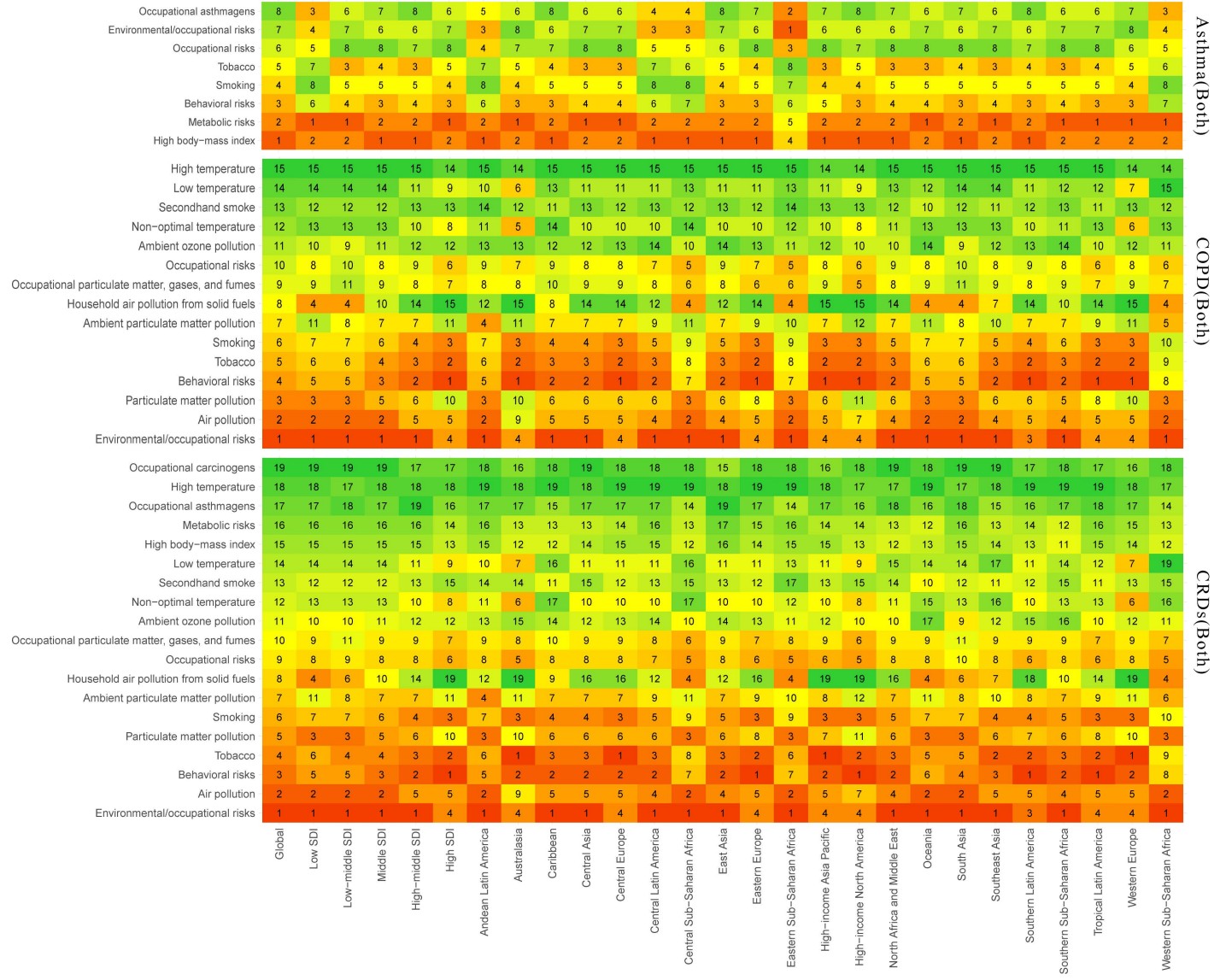

**Fig 9. Risk factors associated with the DALYs Rate of CRDs, COPD, and Asthma at the global level, 5 SDI regions, and 21 GBD regions.**

comparison and analyzes the before-and-after differences within regions based on the period-cohort effect. We not only describe the burden trends before and after the pandemic but also delve into the underlying driving factors, taking into account the profound impacts of population-gender structure, geographical regions, historical backgrounds, and economic income. Based on the findings of this study, we observe that at least by 2021, and perhaps driven by the pandemic, global CRD prevention and control efforts appeared to be steadily improving, yet hidden risks remain. Therefore, compared to previous studies, this research holds significant warning implications.

## The "gradually aging Earth"

Analysis of the GBD database shows that COPD is the main driver of the overall burden from CRDs, followed by asthma and, lastly, ILD & PS. Over the past three decades, the global CRD burden has generally declined. However, except for

asthma, both COPD and ILD & PS have continued to rise. Among people aged 55 years and older, COPD accounts for almost 80 percent of the total COPD burden across all ages, and ILD & PS displays the same pattern. Additionally, the burden of all CRDs-related risks increases with age and is strongly associated with socioeconomic status. This is mainly reflected in the local drift, where the stabilization age interval shifts earlier as SDI decreases, implying that in low SDI regions, the increasing burden among older adults will occur earlier. This phenomenon is associated with unfavorable healthcare conditions in low-income regions and can be alleviated as regional health levels improve, which is considered a natural trend.

Most critically, we found that the CRD burden examined in this study showed abnormal variations across different regions. Specifically, high SDI regions had the fastest increase in incidence, while prevalence continued to rise in low SDI regions. This indicates that the health of the elderly population in high SDI regions is worsening, while the proportion of elderly individuals in low SDI regions continues to grow, heralding a rapid global march towards an aging population framework. The main reason behind this phenomenon can be attributed to the sustained and stable development across regions globally over the past 30 years, with China serving as a prime example. In the 75 years since the establishment of modern China after World War II, influenced by policies such as reform and opening-up and the one-child policy, the country has transitioned from a period of rapid population growth to a decline in birth rates and an ongoing shift toward an aging population structure. In 2018, the decline in the working-age population marked the end of China's demographic dividend, leading to a slowdown in economic growth. In present-day China, the era of the one-child generation has arrived, and each individual family is now responsible for supporting more elderly members than in the past, which implies a heavier economic burden and potentially less access for the elderly to timely health checkups. Since 2017, the rate of global population growth has slowed, and the population age structure has generally tilted towards aging, particularly with the shift in the population structure from youth to middle age in low- and middle-income regions. These countries will become a microcosm of China's developmental trajectory [9]. Meanwhile, in 2021, fertility rates worldwide were declining, and not a single country reached the replacement level. The resulting population shrinkage and aging will place enormous pressure on healthcare systems, social security, and the labor force [19]. European and North American countries have long been in the aging stage of population structure [27,28], but factors such as political ideology—like the anti-abortion laws in the United States—have caused fertility rates in these regions to remain low, driving these countries further along the path of aging. Studies have shown that repeated abortions increase health risks and impair fertility [29,30]. Measures like sex education and contraceptives, which were originally meant to be promoted in low-income countries to control explosive population growth, are now being misused in high-income countries, thereby harming societal welfare. In the future, more newborns will be born in countries in low SDI regions [19], and these regions are likely to become the major sources of the global CRD burden. However, for now, these regions are still in a phase characterized by high incidence, high mortality, and high DALYs. This demographic shift directly contributes to health inequality in COPD and ILD&PS among older adults in high-income regions.

Certain CRDs have significant effects on specific populations. Overall, the respiratory disease burden caused by environmental particulate pollution or smoking shows no significant gender difference among older adults. However, smoking increases the risk of COPD and ILD&PS in elderly men more than in women. Nevertheless, after excluding the influence of smoking habits and population size, women are more likely than men to suffer impaired lung function due to smoking or particulate pollution [31]. This is because, especially in developing countries, women are primarily responsible for activities such as heating and cooking, resulting in a significantly higher exposure rate compared to men [32], compounded by physiological small airway vulnerability and chronic inflammation [33]. Additionally, the asthma burden is significantly higher in older women compared to men, which can be attributed to changes in hormone levels that influence lipid metabolism and immune responses, ultimately leading to differences in asthma prevalence between men and women. This also explains why a higher BMI poses a greater asthma risk for women [34].

## CRDs in the wake of the pandemic

Undeniably, the COVID-19 pandemic has had a significant impact on the prevention and control of CRDs globally, and at the same time, the global health landscape has already shifted [1]. During the period from 2019 to 2021, the overall outlook for CRDs was concerning, with an increasing trend in the burden of disease globally. This was especially true for asthma, followed by COPD, both of which are the primary drivers of CRD-related burden. All CRDs discussed in this study showed similar results, with the overall burden particularly severe in Asia and North America, while middle- and low-SDI regions faced higher risks of disability and mortality. Furthermore, aging itself is a prominent risk factor for severe illness and mortality due to COVID-19 [35], which increases the prevalence risk of CRDs among older adults.

This shift has two main causes. First, the high population density in Asia [36], particularly in China, made it the first region hit by the COVID-19 outbreak, which then rapidly spread across Asia. Second, middle- and low-income regions [37], particularly India [38], faced inefficient lockdown strategies and had inadequate healthcare infrastructure, making these areas severely affected and leading to an increasing absolute burden of CRDs in South Asia. The overall risk of CRD prevalence has always been high in high-income regions, but this does not necessarily mean they were the most severely affected by the pandemic, as these areas have an aging population structure. However, there are exceptions, such as the United States. As of 2022, the United States had over 80 million infections and more than 1 million deaths. This was primarily because the federal government downplayed the severity of COVID-19 and mishandled the early stages of the pandemic [39], but due to the availability of high-quality healthcare, the risk of disability and death was relatively low [37].

Compared to the burden changes in COPD and ILD&PS during the pandemic, asthma prevalence and incidence increased rapidly. On the one hand, this could be because asthma exacerbations prompted more urgent medical attention. On the other hand, prolonged lockdowns, such as those in China, increased the prevalence risk of CRDs among older adults. The extended lockdowns forced individuals with allergies to remain exposed to indoor pollutants for longer periods, including pet dander, toxic gases, and particulates from solid fuel or cooking [40]. Although outdoor air pollution significantly improved, the overall population exposure risk to indoor pollution increased, especially concerning $NO_2$ [41]. Moreover, studies have shown that daily smoking rates increased during lockdowns [42], further raising health risks for elderly individuals at home.

## The presence of nations in regional changes

India, being vast and resource-rich, was expected to have better pandemic control outcomes compared to its neighboring country, Pakistan. However, in reality, India's testing effectiveness was poor, case numbers continued to rise, and it was hit by the most severe pandemic impact. In contrast, in Pakistan, the government implemented smart lockdowns and made effective use of resources, thereby minimizing the health crisis for the general public [38]. Therefore, the disease burden of a country cannot be evaluated solely from the perspective of SDI. Geographic regions, climate change, historical background, government strategies, and cultural traditions all have profound effects on the overall burden of CRDs among the elderly in that region.

Regardless of COPD, asthma, or ILD&PS, the results of the cohort effect reflect the deep implications of regional differences. On the one hand, for individuals born in high-income countries during the mid-20th century, the overall risk of CRD prevalence was significantly higher compared to those born later. However, in low-income regions, the situation was quite the opposite, with those born earlier experiencing stable prevalence risk that then gradually increased over time. This phenomenon does not imply that the healthcare conditions in high-income countries like Germany and the United Kingdom were poor. On the contrary, it reflects the sparse population and lack of epidemiological data in low-income countries, and it also means that only the fittest individuals survived into this century to be surveyed. This interpretation aligns with the unique case observed in high-income countries, where the prevalence risk of various CRDs in the United States during the past century was relatively minimal compared to other developed countries. Since the 19th century, the

U.S. embraced the Monroe Doctrine, remained untouched by wars on its mainland, experienced rapid economic growth, and led the world in healthcare and medical industries. By contrast, Japan, China, Russia, and Germany all experienced varying levels of cohort effects for asthma, COPD, and ILD&PS, largely because they suffered devastating impacts during World War II. On the other hand, cohort effects indicate that, with time, the overall burden of CRDs in middle- and low-income countries is on the rise. This trend is also observed in upper-middle-income countries, though the impact is less pronounced compared to the earlier periods.

After the war, nations needed to rebuild and develop. People born during this period were affected by varying degrees of CRDs, particularly asthma. Since World War II, the chemical industry has rapidly expanded, with approximately 1,500 new chemicals being released annually [43]. At the same time, the pace of global industrialization increased, and diesel locomotives became widely used [44], resulting in the emission of large quantities of sulfur dioxide, nitrogen oxides, particulate matter, and carbon monoxide into the atmosphere. Populations were extensively exposed to CRD-related risk factors, laying hidden risks for a surge in CRDs in their middle and older years.

From the Cold War between the United States and the Soviet Union to the collapse of the Soviet Union, the USSR experienced severe political and economic turmoil. In the latter half of the 20th century, the Soviet Union was overly reliant on heavy industry and military industry, leading to serious air pollution problems, while nuclear leaks further exacerbated water and soil contamination. An economic crisis also emerged in the late 1970s. These combined factors may be one of the reasons why the burden of ILD&PS in Russia followed a different trend compared to other countries. Regional extreme climates also had an impact on Russia, particularly extreme low temperatures. In most areas, cold effects were predominant, and cold weather posed a greater disease burden compared to hot weather [45,46], especially affecting elderly men [47]. Research also indicates that ILD&PS often acutely worsens during winter [48]. As the trend of global warming gradually takes shape and Russia's economy recovers, the "Belt and Road" initiative has significantly contributed to economic development in Eastern Europe and has had a positive impact on healthcare outcomes [49]. Together, these factors have helped to reduce the burden of ILD&PS in Russia.

These past issues have now taken on new forms. With the acceleration of urbanization, new environmental pollution problems have emerged, such as the increased presence of phthalates in the environment [50], indoor dust mite problems [51], and the ongoing harm from smoking. In high-income regions, overweight and obesity are among the key factors affecting respiratory health in older adults [52]. In the United Kingdom, over 60% of adults are overweight or obese, while in the United States, this figure reaches 80%. Additionally, since the 21st century, extreme weather events have become increasingly frequent, and global warming, along with the urban heat island effect, has led to a continuous increase in the overall burden of COPD among older adults. This has had a sustained impact on hot regions, including Southeast Asia, Sub-Saharan Africa, North Africa, and the Middle East [46,53].

The advent of the post-pandemic era and ongoing global demographic shifts necessitate heightened focus on the evolving burden of CRDs among the elderly population. COPD, a primary driver of CRDs in middle-aged and older adults, is projected to experience a modest increase in global incidence according to predictive models. However, the widening confidence intervals associated with longer prediction periods indicate greater uncertainty in these outcomes. While asthma prevalence overall shows a declining trend over a 30-year horizon, this trajectory was reversed in most regions due to the impact of the pandemic. Concurrently, as global regions progressively downplay pandemic influences, the cyclical resurgence of the virus and its variants has contributed to an overall decline in respiratory health among younger populations, potentially predisposing them to adverse respiratory outcomes in later life. ILD&PS exhibited trends counter to the overall pattern during the pandemic, yet this does not definitively indicate a reduced disease burden. Increased time spent indoors during isolation likely led to underdiagnosis, and fatalities often attributed solely to COVID-19 infection may have obscured the true mortality impact of ILD&PS. The consequences of SARS-CoV-2 infection are persistent; studies indicate that 12% of patients exhibit fibrotic changes one year post-infection [54], with individuals experiencing recurrent infections potentially facing more significant long-term effects.

## Limitations of this study

All data in this study come from the GBD database, and the resulting bias inevitably affects our findings. First, the GBD depends heavily on national and regional reporting. Data quality is poor in many low- and middle-income countries, and diagnostic criteria vary, especially for hard-to-detect diseases such as ILD & PS. Second, many countries lack effective death-registration systems and must rely on verbal reports to estimate mortality. Such methods cannot precisely capture CRD-related deaths. Third, important risk factors—such as the number of COVID-19 infections, baseline diseases in older adults, and genetic information—are not available in the GBD, masking their influence. Fourth, the GBD applies extensive statistical modeling and estimation, particularly where direct data are missing. Although these methods have been rigorously validated, they still rest on assumptions and projections. Fifth, because environmental conditions and disease-control policies differ across countries, we could not evaluate every region. Global trends and risk forecasts, therefore, may not reflect a specific nation's trajectory. Finally, projections based on the BAPC model do not consider future changes in CRD-related policies or unforeseen public-health events, so the future course of CRDs remains uncertain. The ultimate burden will depend on how effectively we manage these diseases.

## Conclusion

CRDs represent a major public health issue, imposing significant health and economic burdens on older adults. By accounting for the world's age structure, this study focused on their absolute burden in this population. Over the past 30 years, despite the overall burden of CRDs showing a declining trend, this is mainly attributed to the effective control of asthma. However, under the impact of the pandemic, the overall burden of asthma has significantly shifted worldwide, causing ongoing harm to older adults, alongside COPD. The analysis results from the APC model serve as a wake-up call: during the pandemic, incidence rates in high-income countries have risen rapidly, while prevalence rates in low-income countries have been accumulating quickly. We urgently need to identify the causes of these trends and reverse them. Furthermore, considering the irreversible respiratory damage that aggressive global development strategies in the last century have inflicted on the previous generation, we must strike a balance between development and environmental protection in this century. High-income countries should not use this as a reason to inhibit the development of middle- and low-income countries, because development often comes with sacrifices. Instead, we must harness global knowledge to minimize harm to humanity, enabling more people to experience a happy and healthy life free from CRDs as they age.

## Supporting information

**S1 File.  Methodological details of the Global Burden of Disease (GBD) database and analytical approaches employed in this study.**
(PDF)

**S2 File.  Complete supporting results derived from the GBD data analysis conducted in this investigation.**
(PDF)

## Acknowledgments

We would like to express our gratitude to Longhua Hospital, affiliated with Shanghai University of Traditional Chinese Medicine, and Director Hong Fang for their support of this study.

## Author contributions

**Conceptualization:** Anran Xu, Shaobin Li, Chengyan Zhan, Hong Fang.
**Data curation:** Anran Xu, Yinqin Liu, Chengyan Zhan, Yanqi Cheng, Hong Fang.

**Formal analysis:** Anran Xu, Yinqin Liu, Chengyan Zhan.

**Funding acquisition:** Anran Xu, Yanqi Cheng, Hong Fang.

**Investigation:** Anran Xu, Yinqin Liu, Shaobin Li, Donghua Zhou.

**Methodology:** Anran Xu.

**Project administration:** Anran Xu, Shaobin Li.

**Resources:** Anran Xu, Yinqin Liu, Yanqi Cheng, Chen Zhang, Hong Fang, Donghua Zhou.

**Software:** Anran Xu, Yinqin Liu, Chengyan Zhan, Yanqi Cheng, Chen Zhang.

**Supervision:** Anran Xu, Hong Fang.

**Validation:** Anran Xu, Shaobin Li, Hong Fang, Donghua Zhou.

**Visualization:** Anran Xu, Yinqin Liu, Chengyan Zhan, Yanqi Cheng, Chen Zhang.

**Writing – original draft:** Anran Xu, Shaobin Li, Yanqi Cheng, Hong Fang.

**Writing – review & editing:** Anran Xu, Chen Zhang, Hong Fang.

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
