## [Decision Letter · Decision Letter 0]

PONE-D-25-18984Global Burden of Major Chronic Respiratory Diseases Among Older Adults Aged 55 and Above from 1990 to 2021: Changes, Challenges, and Predictions Amid the PandemicPLOS ONE

Dear Dr. Fang,

Thank you for submitting your manuscript to PLOS ONE. After careful consideration, we feel that it has merit but does not fully meet PLOS ONE’s publication criteria as it currently stands. Therefore, we invite you to submit a revised version of the manuscript that addresses the points raised during the review process.

We look forward to receiving your revised manuscript.

Kind regards,

Alireza Sadeghi, M.D., M.P.H.

Academic Editor

PLOS ONE

Journal Requirements:

5. Please amend your list of authors on the manuscript to ensure that each author is linked to an affiliation. Authors’ affiliations should reflect the institution where the work was done (if authors moved subsequently, you can also list the new affiliation stating “current affiliation:….” as necessary).

6. Please amend either the abstract on the online submission form (via Edit Submission) or the abstract in the manuscript so that they are identical.

7. We note that Figure 5 in your submission contain [map/satellite] images which may be copyrighted. All PLOS content is published under the Creative Commons Attribution License (CC BY 4.0), which means that the manuscript, images, and Supporting Information files will be freely available online, and any third party is permitted to access, download, copy, distribute, and use these materials in any way, even commercially, with proper attribution. For these reasons, we cannot publish previously copyrighted maps or satellite images created using proprietary data, such as Google software (Google Maps, Street View, and Earth). For more information, see our copyright guidelines: http://journals.plos.org/plosone/s/licenses-and-copyright.

a. You may seek permission from the original copyright holder of Figure 5 to publish the content specifically under the CC BY 4.0 license.

8. Please upload a new copy of Figure 1, 2, 3, 4, 5, 6, 7, 8, and 9 as the detail is not clear. Please follow the link for more information: https://blogs.plos.org/plos/2019/06/looking-good-tips-for-creating-your-plos-figures-graphics/

9. Please remove all personal information, ensure that the data shared are in accordance with participant consent, and re-upload a fully anonymized data set.

10. Please include captions for your Supporting Information files at the end of your manuscript, and update any in-text citations to match accordingly. Please see our Supporting Information guidelines for more information: http://journals.plos.org/plosone/s/supporting-information.

Additional Editor Comments :

Dear Authors,

Thank you for submitting your manuscript for consideration. Your work presents a comprehensive analysis of the GBD data, populated with different analyses. The study is well-structured (but too lengthy), and you have done an excellent job in discussing the limitations of your analysis as well as the policy implications of your findings. However, to further improve the clarity, rigor, and impact of your manuscript, I have a some suggestions for revision.

1. Clarity and Readability

The current writing can be challenging to follow in many sections. I recommend revising for conciseness and clarity, ensuring that each sentence conveys its point as directly as possible. Consider breaking down long, complex sentences.

The abstract and results sections are overly detailed and could benefit from restructuring to focus on the key objectives, methods, findings, and implications in a more succinct manner. A standard abstract typically follows the Background, Methods, Results, Conclusions framework in 200-250 words. Remember to use your keywords in your title and abstract strategically to improve indexing and findability.

2. Abbreviations

Several abbreviations (e.g., COVID-19, GBD, ILD&PS) are used without first being defined in the main text. Please ensure that all abbreviations are spelled out in full at first mention, followed by the abbreviated form in parentheses (e.g., Global Burden of Disease (GBD)). This is essential for readability, especially for readers outside your immediate field.

3. Citations and Literature Context

To strengthen the scholarly foundation of your work, please ensure that you cite and discuss the most relevant and recent literature in your field. This includes studies that have used similar methodologies or addressed related research questions, even in other countries and regions (if you don't believe you should cite them, provide them in your response so the reviewers can compare your work with similar works).

4. Figures and Tables

The image quality in the current version is suboptimal. If this is due to PDF rendering issues, please ensure high-resolution figures are submitted elsewhere or hosted on the web with links. If the original figures are of low resolution, consider regenerating them for better clarity. If possible, please provide vectorized images.

Figure formatting: As a general rule, plot titles should be moved to the figure captions rather than embedded in the plots themselves (e.g., Figure 4). This improves consistency with journal formatting standards.

Results section: While detailed, this section could be more concise. Avoid restating exact numerical values and facts that are already clearly presented in tables/figures—instead, focus on trends, key findings, and their significance.

5. Reviewer Comments

Please ensure that all points raised by the reviewers are addressed in your revision. If you disagree with any suggestions, provide a well-justified response explaining your reasoning.

Final Remarks

Your manuscript has strong potential, and with these refinements, it will make an even more valuable contribution to the literature. I appreciate the effort you have put into this work and look forward to reviewing your revised submission.

Reviewers' comments:

Reviewer's Responses to Questions

**Comments to the Author**

1. Is the manuscript technically sound, and do the data support the conclusions?

Reviewer #1: Yes

Reviewer #2: Partly

2. Has the statistical analysis been performed appropriately and rigorously? 

Reviewer #1: Yes

Reviewer #2: Yes

3. Have the authors made all data underlying the findings in their manuscript fully available?

Reviewer #1: Yes

Reviewer #2: Yes

4. Is the manuscript presented in an intelligible fashion and written in standard English?

Reviewer #1: Yes

Reviewer #2: Yes

5. Review Comments to the Author

Reviewer #1: Thank you for the submission and it was really a pleasure to read such a powerful and informative study. I have very minor suggestions for you to address.

Abstract: We recommend to write abstract not exceeding 300 words. That is why I would suggest you to concise the abstract a little more.

Introduction:

There is one sentence I found is long and dense. Breaking them into shorter sentences will improve readability for non-professionals too:

Rather than writing, "The improvement in global health status over the past three decades has proven fragile, vanishing under the impact of the pandemic, leading to a reversal and increase in the overall disease burden."

I would suggest to write, "Although global health improved over the past three decades, the COVID-19 pandemic reversed much of that progress, increasing the overall disease burden." You do not need to copy and paste, if you can simplify with your own sentence, you are most welcome.

Abbreviations: Define abbreviations upon first appearance in the text, I think that is enough. It is not necessary to put it with a different subheading.

Acknowledgement: Please acknowledge the Research Assistants, Data collectors or other contributors too who did not meet the eligibility criteria of authorship (if any).

Funding/Financial Disclosure: So, I am confused here. In the submission system, you have mentioned that the study was not funded but in the manuscript, you wrote, "This study was supported by Longhua Hospital, affiliated with Shanghai University of Traditional Chinese Medicine, and the Shanghai Municipal Fund." Kindly mention the clear funding statement in the submission system, if the study was really funded or not. If it was funded then which section was funded, study design, fieldwork, data analysis, decision to publish or preparation? Kindly mention and clarify the section.

Kindly address the above mentioned issues. Good Luck.

Reviewer #2: Thank you for the opportunity to review this important manuscript exploring the global burden of chronic respiratory diseases (CRDs) among adults aged 55 and above using GBD 2021 data and multiple modeling approaches. The study addresses a globally significant issue and utilizes large-scale epidemiological data; however, major revisions are necessary to ensure clarity, methodological transparency, interpretive accuracy, and compliance with PLOS ONE's publication standards. Below are my specific concerns and suggestions:

1. Methodological Clarity and Transparency

The manuscript uses several complex models (e.g., Joinpoint regression, Age-Period-Cohort [APC], Bayesian APC [BAPC]) but lacks adequate explanation of how these models were selected, validated, and interpreted. Provide a supplementary file describing model specifications, priors, convergence checks (e.g., trace plots or R-hat values), and reasoning for using multiple models rather than selecting one consistent analytical approach.

2. Overinterpretation of Projections and Trends

The manuscript presents future CRD projections with unwarranted certainty, lacking discussion of key assumptions or potential global disruptions. The authors should temper predictive claims, clearly acknowledge model limitations, and consider uncertainty factors such as post-pandemic health shifts, tobacco trends, or environmental policy changes.

3. Presentation and Structure of Results

While the dataset is extensive, the manuscript presents too much detail without clear prioritization of statistically or clinically meaningful findings. Streamline the results to focus on key patterns (e.g., high-burden regions, divergent trends by SDI quintile). Suppress or summarize non-significant results and consolidate overlapping figures/tables for clarity.

4. Language and Expression

The manuscript has several grammatical errors and inconsistent phrasing, particularly in the Introduction and Discussion, which sometimes obscures the intended meaning, e.g. “This implies that while addressing environmental issues, it is also crucial to consider other epidemiological shifts” is vague and lacks subject clarity. It is unclear what "this" refers to or what the “other shifts” are. Revise for grammatical correctness and precision, and consider a professional language editing.

5. Ethical and Data Transparency

Although the study uses publicly available data, ethical oversight and data/code availability are not adequately addressed. In the Ethics section, “Not applicable” is stated without explanation. Additionally, while GBD is cited, the manuscript does not share processed data tables or scripts used for analysis. Provide a brief justification for the ethical exemption (e.g., use of secondary, de-identified data) and comply with PLOS ONE’s reproducibility standards by depositing analysis scripts (e.g., R code for APC/BAPC models) and processed data tables in a public repository.

This manuscript covers an important global health topic and utilizes reputable data sources, but significant revisions are required to meet publication standards. I encourage the authors to: justify and clarify their modeling approaches adequately, interpret projections with caution, improve visual and textual presentation of results, revise language for clarity, and provide appropriate transparency on data ethics and availability.

With these revisions, the study will be more methodologically sound, interpretable, and useful to the global health and policy community.

6. PLOS authors have the option to publish the peer review history of their article (what does this mean? ). If published, this will include your full peer review and any attached files.

**Do you want your identity to be public for this peer review?** For information about this choice, including consent withdrawal, please see our Privacy Policy .

Reviewer #1: No

Reviewer #2: No

---

## [Author Response · Author response to Decision Letter 1]

1 Jul 2025

Response to Reviewers – Point-by-Point Reply

Dear Editor and Reviewers,

We sincerely appreciate the time you have taken to review our manuscript. We have carefully considered each of the reviewers' comments and have addressed them thoroughly, with major changes highlighted within the manuscript text. We sincerely regret the delay of nearly one month in submitting this revision, which was primarily due to extensive modifications made to the study's content. Overall, the Abstract, Methods, and Results sections have been comprehensively revised and streamlined. Additionally, we have provided a supplementary methodological appendix and implemented a new analytical approach (decomposition analysis) to enhance the clarity of our findings. Finally, below is our point-by-point response to the journal requirements, the Editor's comments, and the reviewers' suggestions.

Response to the Journal 1

Response to the Editor 3

Response to the Reviewers 6

Reviewer #1 6

Reviewer #2: 8

Response to the Journal

Response: The manuscript formatting has been revised in accordance with established research conventions and journal guidelines. Table and figure captions have been modified and are now positioned following the reference list to enhance readability of the main text.

2. PLOS requires an ORCID iD for the corresponding author in Editorial Manager on papers submitted after December 6th, 2016. Please ensure that you have an ORCID iD and that it is validated in Editorial Manager.

Response: Both the first author and corresponding author have registered and activated ORCID iDs.

3. Please note that PLOS ONE has specific guidelines on code sharing for submissions in which author-generated code underpins the findings in the manuscript. In these cases, we expect all author-generated code to be made available without restrictions upon publication of the work.

Response: The raw data and analysis code have been deposited in the Open Science Framework (OSF) repository (https://osf.io/4aq67/?view_only=0db1dec18bc04d4ebc4ae2ee29f8d098). These materials contain no identifiers or personal information. They are accessible online and available for download (if file size permits).

Response: The funding information has been updated with the correct grant numbers, now included at the end of the main text.

5. Please amend your list of authors on the manuscript to ensure that each author is linked to an affiliation. Authors’ affiliations should reflect the institution where the work was done

Response: Author affiliations have been verified and corrected where necessary.

6. Please amend either the abstract on the online submission form (via Edit Submission) or the abstract in the manuscript so that they are identical.

Response: The online abstract has been comprehensively revised to ensure full consistency with the manuscript content.

7. We note that Figure 5 in your submission contain [map/satellite] images which may be copyrighted. All PLOS content is published under the Creative Commons Attribution License (CC BY 4.0), which means that the manuscript, images, and Supporting Information files will be freely available online, and any third party is permitted to access, download, copy, distribute, and use these materials in any way, even commercially, with proper attribution. For these reasons, we cannot publish previously copyrighted maps or satellite images created using proprietary data, such as Google software (Google Maps, Street View, and Earth).

Response: Maps were generated using publicly available R packages (ggmap and maps), which utilize open-source cartographic data from the Natural Earth project (https://www.rdocumentation.org/packages/maps/versions/3.4.2). Package maintainers have documented this data provenance. Relevant methodological details appear in the main text.

8. Please upload a new copy of Figure 1, 2, 3, 4, 5, 6, 7, 8, and 9 as the detail is not clear.

Response: All figures in the main manuscript have been repositioned immediately following the References section and re-uploaded. Figures for supplementary files have been separately re-uploaded. Image formats were converted to meet journal specifications for resolution and clarity, with verification that all figures are scalable and retain sufficient resolution to examine intricate details upon magnification.

9. Please remove all personal information, ensure that the data shared are in accordance with participant consent, and re-upload a fully anonymized data set.

Response: No personal information is present in either the OSF repository or supplementary files.

10. Please include captions for your Supporting Information files at the end of your manuscript, and update any in-text citations to match accordingly.

Response: Supporting information statements have been added to the end of the main text, with non-essential content removed per journal specifications.

Response to the Editor

1. Clarity and Readability

The current writing can be challenging to follow in many sections. I recommend revising for conciseness and clarity, ensuring that each sentence conveys its point as directly as possible. Consider breaking down long, complex sentences.

The abstract and results sections are overly detailed and could benefit from restructuring to focus on the key objectives, methods, findings, and implications in a more succinct manner. A standard abstract typically follows the Background, Methods, Results, Conclusions framework in 200-250 words. Remember to use your keywords in your title and abstract strategically to improve indexing and findability.

Response: After incorporating the suggestions from you and the other reviewers, we have revised and streamlined a substantial amount of material in the abstract and in the introduction, methods, and results sections of the main text, greatly optimizing the overall length of the manuscript. We have also refined sentence phrasing to enhance readability. All major changes are highlighted in the revised manuscript. In the Results section, for each disease we interpret the findings from three levels: (1) a comprehensive description of the current burden and overall trend; (2) an analysis of the impact of the COVID-19 pandemic on that burden; and (3) an assessment of health inequalities, the effects of population aging, and possible future trajectories.

2. Abbreviations

Several abbreviations (e.g., COVID-19, GBD, ILD&PS) are used without first being defined in the main text. Please ensure that all abbreviations are spelled out in full at first mention, followed by the abbreviated form in parentheses (e.g., Global Burden of Disease (GBD)). This is essential for readability, especially for readers outside your immediate field.

Response: We have standardized all abbreviations. In the abstract, each abbreviation is defined in full upon first use. In the main text, starting from the Introduction, every abbreviation’s first occurrence is clearly defined.

3. Citations and Literature Context

To strengthen the scholarly foundation of your work, please ensure that you cite and discuss the most relevant and recent literature in your field. This includes studies that have used similar methodologies or addressed related research questions, even in other countries and regions (if you don't believe you should cite them, provide them in your response so the reviewers can compare your work with similar works).

Response: We have replaced a few references in the main text to ensure they reflect the most recent publications. Many of these studies derive from analyses of the GBD database, particularly pre-pandemic CRD research. In addition, numerous citations come from the GBD Collaborators, who are official members of the GBD consortium; their work is rigorously peer-reviewed internally by the GBD organization. We are pleased to have been approved as junior collaborators with the GBD database. Furthermore, we have provided additional methodological details in “Supplementary File 1,” which includes 24 methodological references to facilitate comparison by other researchers. You and the other reviewers can consult both the main text and Supplementary File 1 for these details.

4. Figures and Tables

The image quality in the current version is suboptimal. If this is due to PDF rendering issues, please ensure high-resolution figures are submitted elsewhere or hosted on the web with links. If the original figures are of low resolution, consider regenerating them for better clarity. If possible, please provide vectorized images.

Figure formatting: As a general rule, plot titles should be moved to the figure captions rather than embedded in the plots themselves (e.g., Figure 4). This improves consistency with journal formatting standards.

Response: In accordance with the journal’s requirements and your suggestion, we have reformatted all figures and re-uploaded them as a single package. Figures and tables now appear after the References to improve the flow of the main text while meeting journal standards. Supplementary figures have also been compressed and re-packaged. All figure captions have been cleaned up, and only the images themselves are retained to comply with format guidelines. We have confirmed that every figure can be resized and viewed in detail.

4.1 Results section: While detailed, this section could be more concise. Avoid restating exact numerical values and facts that are already clearly presented in tables/figures—instead, focus on trends, key findings, and their significance.

Response: We noted that there was some redundancy in the Results, and have made extensive revisions. We removed non-essential narrative elements and integrated country- and region-specific patterns into the global three-level framework described above (overall burden and trend; COVID-19 impact; health inequalities, aging effects, and future trends). The specific changes are available for your review in the revised Results section.

5. Reviewer Comments

Please ensure that all points raised by the reviewers are addressed in your revision. If you disagree with any suggestions, provide a well-justified response explaining your reasoning.

Response: Thank you for your attention. We will carefully address each reviewer comment in our detailed point-by-point reply.

Response to the Reviewers

Reviewer #1

1. Abstract: We recommend to write abstract not exceeding 300 words. That is why I would suggest you to concise the abstract a little more.

Response: Thank you for your suggestion. After incorporating your and the editor’s recommendations, we have fully streamlined the abstract, retaining only the most essential content.

2. Introduction:

There is one sentence I found is long and dense. Breaking them into shorter sentences will improve readability for non-professionals too:

Rather than writing, "The improvement in global health status over the past three decades has proven fragile, vanishing under the impact of the pandemic, leading to a reversal and increase in the overall disease burden."

I would suggest to write, "Although global health improved over the past three decades, the COVID-19 pandemic reversed much of that progress, increasing the overall disease burden." You do not need to copy and paste, if you can simplify with your own sentence, you are most welcome.

Response: Following your comments, we have extensively revised the manuscript and condensed its overall length. The Introduction, Methods, and Results sections were the main focus and have been almost entirely reconstructed. In revising the text, we endeavored to enhance readability. For example, we modified the sentence you proposed to read: “Under the impact of the global Corona Virus Disease 2019 (COVID-19) pandemic, the overall disease burden, including CRDs, has reversed and increased.” We hope this meets your expectations. All substantially revised passages have been highlighted; given the volume of changes, there are many highlighted sections, and we trust this is acceptable.

3. Abbreviations: Define abbreviations upon first appearance in the text, I think that is enough. It is not necessary to put it with a different subheading.

Response: We have rewritten the abbreviations section in accordance with your and the editor’s feedback. We removed the unnecessary list of abbreviations that appeared at the end of the original manuscript. In the abstract, full definitions are now provided for all abbreviations. In the main text, each abbreviation is clearly defined at its first occurrence, beginning with the Introduction.

4. Acknowledgement: Please acknowledge the Research Assistants, Data collectors or other contributors too who did not meet the eligibility criteria of authorship (if any).

Response: The acknowledgements have been finalized, and there are no additional parties requiring special acknowledgement.

5. Funding/Financial Disclosure: So, I am confused here. In the submission system, you have mentioned that the study was not funded but in the manuscript, you wrote, "This study was supported by Longhua Hospital, affiliated with Shanghai University of Traditional Chinese Medicine, and the Shanghai Municipal Fund." Kindly mention the clear funding statement in the submission system, if the study was really funded or not. If it was funded then which section was funded, study design, fieldwork, data analysis, decision to publish or preparation? Kindly mention and clarify the section.

Response: We apologize for omitting the funding information in the submission system. In the revised manuscript, we have added the following statement at the end of the main text: “The publication of this study was supported by the Preventive Treatment of Disease in Traditional Chinese Medicine Specialist Alliance Construction Project (Grant No. 2024LM01) of Longhua Hospital affiliated with Shanghai University of Traditional Chinese Medicine.” We will update the submission system to ensure consistency with the revised manuscript.

Reviewer #2:

1. Methodological Clarity and Transparency

The manuscript uses several complex models (e.g., Joinpoint regression, Age-Period-Cohort [APC], Bayesian APC [BAPC]) but lacks adequate explanation of how these models were selected, validated, and interpreted. Provide a supplementary file describing model specifications, priors, convergence checks (e.g., trace plots or R-hat values), and reasoning for using multiple models rather than selecting one consistent analytical approach.

Response: Thank you for pointing out the methodological shortcomings in our previous version. We have undertaken major revisions. The Methods section of the main text now provides appropriate descriptions, and we have added a separate methodological appendix titled Supplementary Material 1. In this appendix we describe in detail all methods used in the present study, including data sources, a comprehensive introduction to the GBD database, and extensive explanations of approaches such as EAPC, APC, and various predictive models. For every method, we discuss both its rationale and its limitations. Full details can be found in Supplementary Material 1.

In GBD research, the complementary use of multiple models is essential and has been widely adopted in earlier studies[1–3]. For example, EAPC averages multiple phases of change and can obscure important turning points. Joinpoint regression, by contrast, objectively identifies statistically significant inflection points and estimates the average annual percent change for each segment. The APC model goes a step further by estimating the independent effects of age, period, and cohort on the burden of chronic non-communicable diseases. Applying APC to individual regions and representative countries enables us to discern differences among birth cohorts across eras and to infer the impacts of environmental conditions, policy shifts, and other historical events. These three methods are therefore indispensable for analyzing overall trends in chronic disease.

Cross-national inequ

---

## [Decision Letter · Decision Letter 1]

Global Burden of Major Chronic Respiratory Diseases Among Older Adults Aged 55 and Above from 1990 to 2021: Changes, Challenges, and Predictions Amid the Pandemic

PONE-D-25-18984R1

Dear Dr. Fang,

We’re pleased to inform you that your manuscript has been judged scientifically suitable for publication and will be formally accepted for publication once it meets all outstanding technical requirements.

Kind regards,

Taiwo Opeyemi Aremu

Academic Editor

PLOS ONE

Additional Editor Comments (optional):

Reviewers' comments:

Reviewer's Responses to Questions

**Comments to the Author**

1. If the authors have adequately addressed your comments raised in a previous round of review and you feel that this manuscript is now acceptable for publication, you may indicate that here to bypass the “Comments to the Author” section, enter your conflict of interest statement in the “Confidential to Editor” section, and submit your "Accept" recommendation.

Reviewer #1: All comments have been addressed

Reviewer #2: All comments have been addressed

2. Is the manuscript technically sound, and do the data support the conclusions?

Reviewer #1: Yes

Reviewer #2: (No Response)

3. Has the statistical analysis been performed appropriately and rigorously? 

Reviewer #1: Yes

Reviewer #2: (No Response)

4. Have the authors made all data underlying the findings in their manuscript fully available?

Reviewer #1: Yes

Reviewer #2: (No Response)

5. Is the manuscript presented in an intelligible fashion and written in standard English?

Reviewer #1: Yes

Reviewer #2: (No Response)

6. Review Comments to the Author

Reviewer #1: No More Changes Required from My Side. I wish you all the best for your future endeavor. Hopefully we will have happy reading.

Reviewer #2: (No Response)

7. PLOS authors have the option to publish the peer review history of their article (what does this mean? ). If published, this will include your full peer review and any attached files.

**Do you want your identity to be public for this peer review?** For information about this choice, including consent withdrawal, please see our Privacy Policy .

Reviewer #1: No

Reviewer #2: **Yes: ** Tinkhani Mbichila

---

## [Editor Report · Acceptance letter]

PONE-D-25-18984R1

PLOS ONE

Dear Dr. Fang,

I'm pleased to inform you that your manuscript has been deemed suitable for publication in PLOS ONE. Congratulations! Your manuscript is now being handed over to our production team.

Kind regards,

on behalf of

Dr. Taiwo Opeyemi Aremu

Academic Editor

PLOS ONE